# The control of transcriptional memory by stable mitotic bookmarking

Maëlle Bellec[1], Jérémy Dufourt[1], George Hunt[2], Hélène Lenden-Hasse[1], Antonio Trullo[1], Amal Zine El Aabidine[1], Marie Lamarque[1], Marissa M. Gaskill[3], Heloïse Faure-Gautron[1], Mattias Mannervik[2], Melissa M. Harrison[3], Jean-Christophe Andrau[1], Cyril Favard[4], Ovidiu Radulescu[5] & Mounia Lagha[1✉]

To maintain cellular identities during development, gene expression profiles must be faithfully propagated through cell generations. The reestablishment of gene expression patterns upon mitotic exit is mediated, in part, by transcription factors (TF) mitotic bookmarking. However, the mechanisms and functions of TF mitotic bookmarking during early embryogenesis remain poorly understood. In this study, taking advantage of the naturally synchronized mitoses of *Drosophila* early embryos, we provide evidence that GAGA pioneer factor (GAF) acts as a stable mitotic bookmarker during zygotic genome activation. We show that, during mitosis, GAF remains associated to a large fraction of its interphase targets, including at *cis*-regulatory sequences of key developmental genes with both active and repressive chromatin signatures. GAF mitotic targets are globally accessible during mitosis and are bookmarked via histone acetylation (H4K8ac). By monitoring the kinetics of transcriptional activation in living embryos, we report that GAF binding establishes competence for rapid activation upon mitotic exit.

[1] Institut de Génétique Moléculaire de Montpellier, University of Montpellier, CNRS-UMR 5535, 1919 Route de Mende, Montpellier 34293 Cedex 5, France. [2] Department of Molecular Biosciences, The Wenner-Gren Institute, Stockholm University, 10691 Stockholm, Sweden. [3] Department of Biomolecular Chemistry, School of Medicine and Public Health, University of Wisconsin-Madison, Madison, WI 53706, USA. [4] Institut de Recherche en Infectiologie de Montpellier, CNRS UMR 9004, University of Montpellier, 1919 Route de Mende, Montpellier 34293 Cedex 5, France. [5] LPHI, UMR CNRS 5235, University of Montpellier, Place E. Bataillon – Bât. 24 cc 107, Montpellier 34095 Cedex 5, France. ✉email: mounia.lagha@igmm.cnrs.fr

Cellular identities are determined by the precise spatio-temporal control of gene expression programs. These programs must be faithfully transmitted during each cellular division. However, with its drastic nuclear reorganization, mitosis represents a major challenge to the propagation of gene expression programs. How cells overcome this mitotic challenge to transmit information to their progeny remains relatively unexplored during embryogenesis[1–3].

Based on live imaging studies and genome-wide profiling experiments on drug-synchronized mitotic cells, it is now well established that a subset of transcription factors (TF), chromatin regulators, and histone modifications are retained on their targets during mitosis[2,4,5]. These TFs can be retained via specific DNA binding, non-specific DNA binding, or a combination of both[5–7].

When the persistence of TF binding during mitosis is associated with a regulatory role in transcriptional activation upon mitotic exit, TFs can be envisaged as mitotic bookmarkers. The kinetics of postmitotic reactivation are often examined by whole-genome profiling experiments of nascent transcription in early G1[8–10]. Combining such approaches with the mitotic depletion of candidate bookmarkers, it was established that some mitotically retained TFs/General TFs/histone marks act as bona fide mitotic bookmarkers[11–13].

Parallel to these multi-omics approaches, imaging of transcription in live cells with signal amplifying systems as the MS2/MCP[14,15] allows for the direct quantification of the kinetics of transcriptional activation upon mitotic exit. With such approaches, mitotic bookmarking has been associated with an accelerated transcriptional reactivation after mitosis in cultured cells[16]. Moreover, this method enabled the visualization of the transmission of active states, referred to as "transcriptional memory" in *Dictyostellium* and in *Drosophila* embryos[17,18]. However, how mitotic bookmarking is associated with the transmission of states across mitosis in the context of a developing embryo remains unclear.

This question is particularly important during the first hours of development of all metazoans, when cellular divisions are rapid and frequent. During this period, there is a substantial chromatin reprogramming and transcriptional activation, called Zygotic Genome Activation (ZGA)[19,20]. The control of this major developmental transition is supervised by key TFs, a subset of which are capable of engaging inaccessible chromatin and foster nucleosome eviction, a defining property of pioneer factors[21–24]. Remarkably, many mitotic bookmarking factors have pioneer factor properties[25].

In *Drosophila melanogaster*, two essential transcription factors with pioneering factor properties, Zelda and GAGA Associated Factor (GAF), orchestrate the reshaping of the genome during ZGA[26–30]. Contrary to Zelda, which is not retained during mitosis and is dispensable for transcriptional memory[31], GAF is known to decorate mitotic chromosomes[28,31,32]. In this study, we asked whether GAF acts as a mitotic bookmarker during ZGA. GAF, encoded by the *Trithorax-like* gene, binds to repeating $(GA)_n$ sequences and displays a broad set of functions including gene activation or silencing, nucleosome remodeling, and chromatin organization[33,34]. In addition, GAF has been shown to be enriched at paused promoters[35,36] and its manipulation in *Drosophila* S2 cells demonstrated a capacity to rapidly evict nucleosomes, thereby facilitating the recruitment of Pol II at promoters[37,38]. Together with its mitotic retention, these properties place GAF as a reasonable candidate for mitotic bookmarking during development.

## Results

### Endogenous GAF is retained during mitosis and stably binds DNA. To investigate the function of GAF during mitosis, we first

characterized its distribution during the cell cycle. With immunostaining, we confirmed that GAF is present on chromatin during all stages of mitosis from prophase to telophase (Fig. 1a)[31,32]. Next, we examined GAF behavior in living embryos using an endogenously GFP-tagged allele of GAF[28] (Fig. 1b and Supplementary Movie 1). During mitosis, a large amount of GAF protein is displaced to the cytoplasm, but a clear pool of GAF protein remains associated with mitotic chromosomes (Fig. 1b).

From both live imaging and immunofluorescence data, we observed a strong GAF signal concentrated in large distinct puncta as well as a more diffuse signal within the nucleus. Consistent with previous work[32], we found that the majority of large GAF puncta are located at the apical side of the nuclei (Supplementary Fig. 1a and Supplementary Movie 2), where at this stage, most of the centromeric heterochromatin is located (Supplementary Fig. 1b)[39]. In contrast to GAF apical foci, the rest of the nuclear space contains a homogeneously distributed GAF signal, potentially representing GAF binding to euchromatin (Supplementary Fig. 1a, b and Supplementary Movie 2). To characterize GAF diffusion and binding kinetics in these regions, we performed Fluorescence Correlation Spectroscopy (FCS) and imaging Fluorescence Recovery After Photobleaching (FRAP)[40] on living GAF-GFP embryos during interphase (Fig. 1c and Supplementary Fig. 1d–h). We could not perform FRAP and FCS during mitosis due to their short duration and rapid nuclear movements[41].

We first performed FCS to characterize fast GAF kinetics (Supplementary Fig. 1d–h). We observed two characteristic times, potentially corresponding to two-diffusion coefficients or to diffusion and a binding reaction. To discriminate between these two scenarios, we performed FCS in the cytoplasm, where binding should not occur. Surprisingly, cytoplasmic FCS revealed two characteristic times, on the same order as those retrieved in the nucleoplasm (Supplementary Fig. 1g, h). Therefore, a two-diffusion component model was used to fit the nucleoplasm autocorrelation curves, giving rise to two apparent characteristic diffusion coefficients (Df) on the order of $22 \, \mu m^2 \, s^{-1}$ and $0.45 \, \mu m^2 \, s^{-1}$ respectively (Supplementary Fig. 1g, h). The fastest Df corresponds to GAF monomer free diffusion, as it falls in the range of diffusion of GFP in cells[42]. The slower diffusion time potentially reflects GAF diffusion within molecular complexes reflecting transient non-specific binding[41,43]. Given this very fast dynamic, we hypothesized that GAF may engage its targets for long timings. To gain access to these longer characteristic times, we performed nuclear FRAP experiments in nc14 *GAF-GFP* embryos, focusing on the middle part of the nuclei (Fig. 1c–f).

As expected for a transcription factor, FRAP recovery curves show more than one unique characteristic time. A reaction-diffusion model was used to fit the recovery curve, and revealed that GAF exhibits two residence times: a short one on the order of seconds (~2 s) that corresponds to an apparent diffusion coefficient of $0.2 \, \mu m^2 \, s^{-1}$ on average, similar to the one observed using FCS, and a longer one on the order of tens of seconds (~58 s) (Fig. 1e, f). We would note here that the fast value of the diffusion coefficient observed with FCS is not experimentally accessible with our FRAP device. A possible interpretation of these two kinetic timescales observed with FRAP experiments would be that the fast residence time corresponds to GAF non-specific binding as observed previously[7], while the long-lived residence time would correspond to sequence-specific binding to its consensus binding sites. Interestingly, similar GAF kinetics were very recently observed in larval hemocytes[44]. We conclude that GAF protein has the intrinsic capacity to stably bind chromatin. This is in sharp contrast to dynamic binding properties recently measured for other transcription factors in the blastoderm embryo as Zelda or Bicoid[31,45,46]. This property

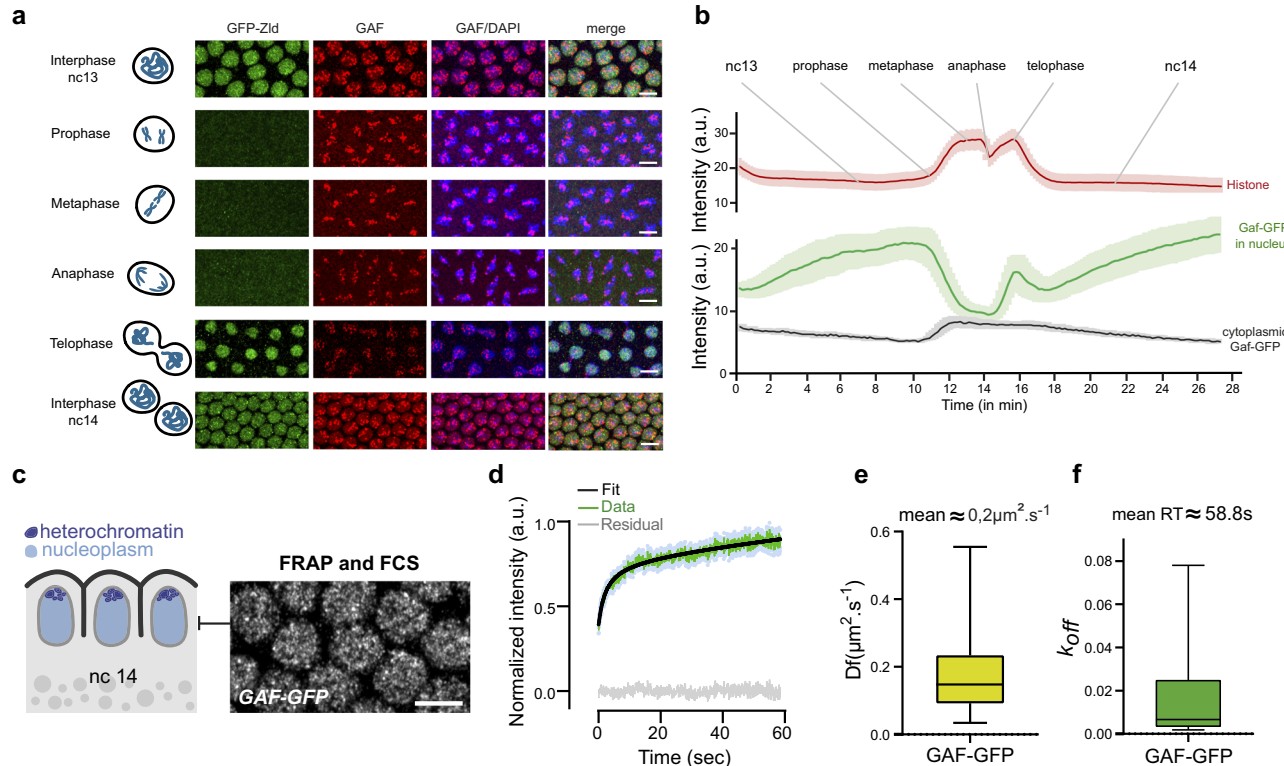

**Fig. 1 GAF dynamics during nuclear cycles and its kinetic properties. a** Maximum intensity projected Z-planes of confocal images from immunostaining of Zelda-GFP (green) and GAF (red) on interphase and mitotic embryos at the indicated stages counterstained with DAPI (blue). The scale bar is 5 μm. **b** Mean fluorescent signal quantifications of GAF-GFP in the nucleoplasm (green) and cytoplasm (gray), and Histone 2 A variant-*mRFP* (shown as "Histone" in the panel) in nucleoplasm during nuclear cycle 13 to 14 extracted from time-lapse movies of embryos expressing GAF-GFP and *His2Av-mRFP* (mean from three movies of three independent embryos). Lighter colors curves represent SEM. **c** Schematic of a sagittal view of nc14 embryos. Nuclei are represented in light blue and apical heterochromatin regions in dark blue. The right panel shows regions targeted by FRAP and FCS, performed on *GAF-GFP* embryos. The scale bar is 5 μm. **d** Mean fluorescence recovery curve (green) from FRAP experiment and fit (black) using a reaction-diffusion model determined at the bleached spot for 23 nuclei from nine nc14 *GAF-GFP* embryos. Light blue dots represent SEM from different nuclei. The gray curve represents the residual of the fit. **e** Estimated diffusion coefficient of GAF-GFP. 23 FRAP traces from 23 nuclei were analyzed. The centered line represents the median and whiskers represent min and max values. **f** Estimated $k_{off}$ (RT: residence time = $1/k_{off}$) of GAF-GFP. 23 FRAP traces from 23 nuclei were analyzed. The centered line represents the median and whiskers represent min and max values.

could be involved in its capacity to associate with mitotic chromosomes during embryonic divisions.

**Capturing GAF mitotic targets genome-wide**. Early *Drosophila* embryogenesis provides an ideal system to study mitosis. Indeed, nuclei of the syncytial embryo divide 13 times synchronously before cellularization[47]. To perform mitotic ChIP, we stained early staged embryos with antibodies against the mitotic specific marker phosphorylation of the serine 10 of the histone 3 (H3S10ph) (Supplementary Fig. 2a)[48,49] and sorted them with a flow cytometer (Fig. 2a and Supplementary Fig. 2b). The pool of embryos were further manually sorted to avoid contamination (Supplementary Fig. 2b). We applied this method to map GAF targets during mitosis and interphase. We retrieved GAF peaks genome-wide in interphase and mitotic samples and classified them into three categories: present only in interphase, only during mitosis, or during both interphase and mitosis, referred to as "mitotically retained" (Fig. 2b–c'). Remarkably, mitotically retained GAF targets represent 37% of interphase targets, corresponding to a group of ~2000 peaks bound by GAF both in interphase and mitotic embryos (Fig. 2b). The mitotically retained loci comprise many key developmental patterning genes, as exemplified by *snail* (*sna*), for which the proximal enhancer shows a GAF mitotic peak (Fig. 2c').

Motif search confirmed that GAF peaks are enriched in GAGAG motifs (Fig. 2d), and are centered inside the reads (Supplementary Fig. 2c). However, this consensus GAF-binding site did not emerge as a significantly enriched motif in the small sample of GAF mitotic-only targets. We, therefore, did not analyze in depth this group of GAF targets. Moreover, there was a substantial degree of overlap (~93.5%) when comparing our interphase GAF peaks with published GAF-ChIP-seq data from bulk 2–4- h embryos[50]. Thus, we established a pipeline able to profile mitotic nuclei at a genomic scale in the absence of drug synchronization.

Interestingly, the number of GAGAG motifs differs between mitotically retained peaks and interphase-only peaks. On average, mitotically retained peaks have 6.2 GAGAG repeats while interphase-only bound targets show 2.9 number of motifs (Fig. 2e). Therefore, we conclude that loci with a significant number of GAF-binding sites are more likely to be bound during mitosis.

Moreover, de novo motif search revealed that while some motifs are present on both categories (interphase only and mitotically retained), a combination of consensus binding sites is specifically enriched in mitotically retained peaks (e.g., dorsal, Supplementary Fig. 2d). GAF mitotically retained targets might therefore be regulated by a distinct *cis*-regulatory logic than those from which GAF dissociates during mitosis.

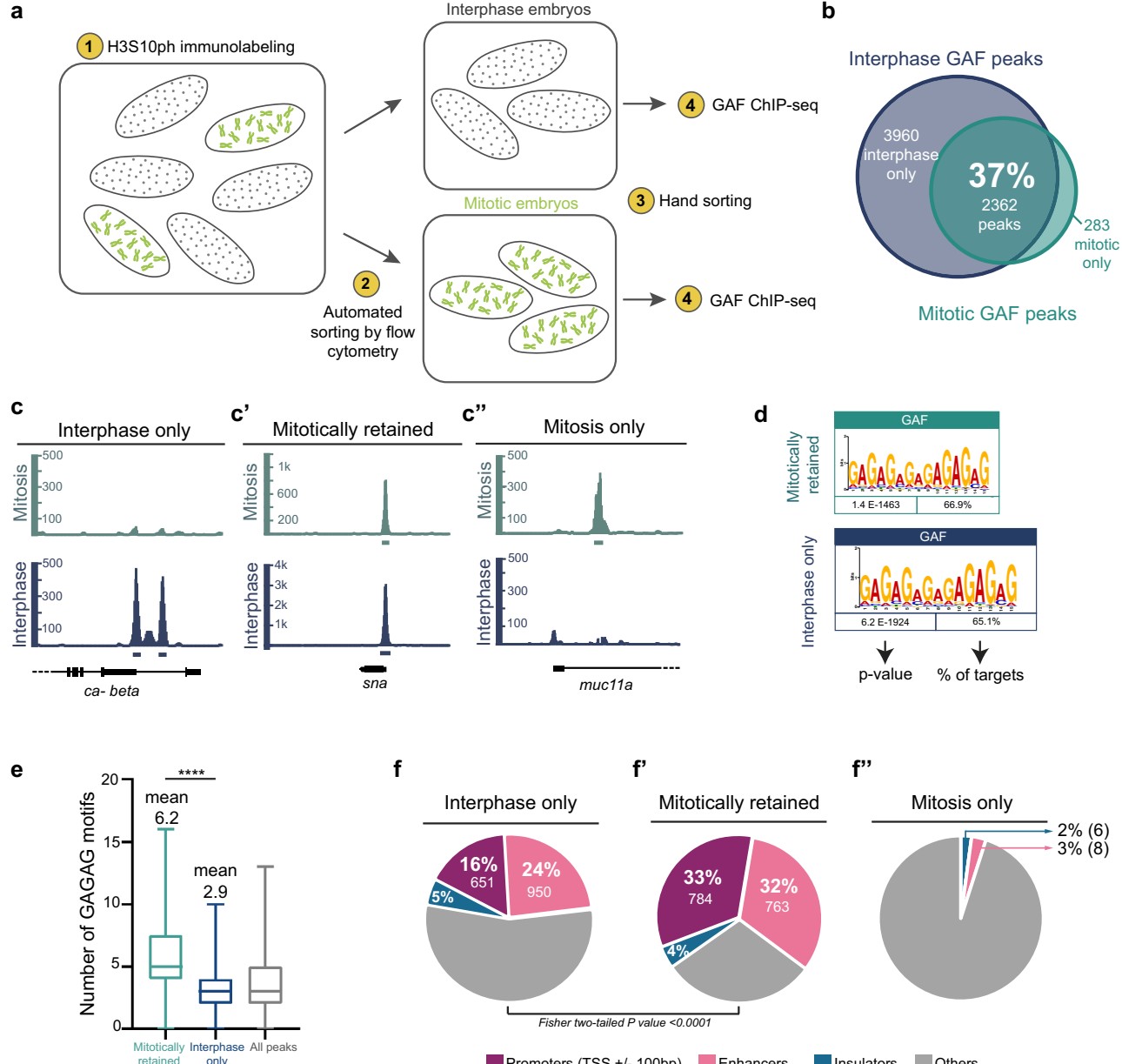

**Fig. 2 Identification of thousands of mitotically retained GAF loci. a** Experimental workflow of mitotic embryo sorting followed by GAF-ChIP-seq. Embryos are immunolabeled with anti-H3S10ph (1), then sorted using a flow cytometer (2). Two pools of embryos are obtained (mitotic and interphase embryos) and are then hand-sorted (3) to remove any contamination. GAF-ChIP sequencing is then performed on approximately 1000 embryos of each condition (4). **b** Venn diagram representing the overlap of called GAF-ChIP-seq peaks between interphase and mitotic embryos. **c, c', c"** Genome browser examples of genes from the identified three categories of GAF-ChIP-seq peaks: interphase only, mitotically retained, and mitosis only, respectively. Rectangles represent the called peaks corresponding to the above profile. **d** (GA)$_n$ motif enrichment within GAF mitotically retained and interphase-only peaks, as reported by MEME. **e** Box plot representing the number of GAGAG motifs within three different classes of GAF peaks: mitotically retained (light blue) $n = 2362$, interphase only (dark blue) $n = 3960$, and all peaks (gray) $n = 283$. The centered horizontal line represents the median; whiskers represent min and max values. Two-tailed Welch's $t$ test ****$P < 0.0001$. **f, f', f"** Proportions of GAF-ChIP-seq peaks that overlap diverse *cis*-regulatory regions in interphase only, mitotically retained and mitosis-only GAF-ChIP-seq.

To better characterize GAF-bound loci, we used existing genomic annotations of *cis*-regulatory modules (enhancers, promoters, and insulators) that were previously obtained from whole-genome profiling of the early *Drosophila* embryo[50–52] or validated via reporter transgenes[53] (see "Methods" and Fig. 2f–f"). This stringent analysis revealed that the majority of GAF mitotically retained regions (65%) correspond to *cis*-regulatory sequences (Fig. 2f). This proportion is higher than the interphase-only peaks (40%, Fig. 2f). A similar clear enrichment of promoters and intronic regions

(probably enhancers) is observed using HOMER tool annotation (Supplementary Fig. 2e).

**Mitotically retained GAF marks accessible regions during ZGA.** As GAF displays pioneering properties in many contexts[28,29,37], we hypothesized that GAF could contribute to chromatin accessibility during mitosis. We, therefore, determined the degree of chromatin accessibility at GAF-bound loci by using

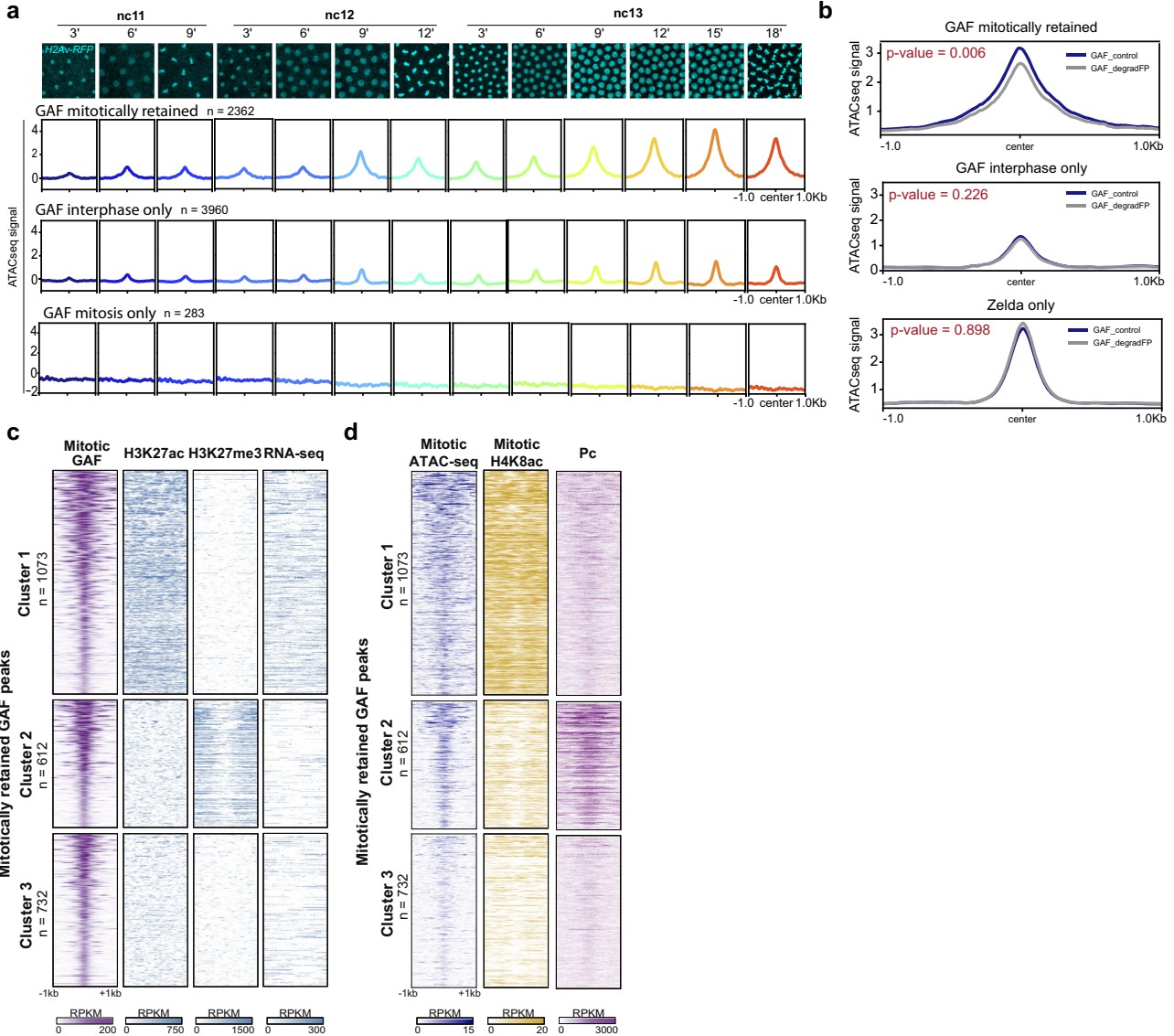

**Fig. 3 Mitotically retained GAF loci become progressively accessible during Zygotic Genome Activation. a** Metagene profiles of ATAC-seq signal[54] centered at mitotically retained interphase only and mitosis-only GAF-ChIP-seq peaks across the indicated stages and represented by the time-lapse images from a movie of *His2Av-mRFP* embryos (cyan). *n*: number of identified GAF peaks. **b** Metagene profiles of ATAC-seq signal in WT (GAF_control, dark blue) and GAF-depleted (GAF_degradFP, gray) embryos (2–2.5 h after egg laying)[28] on GAF mitotically retained, GAF interphase-only and Zelda-only regions. *P* values are from two-sided Wilcoxon rank-sum test with continuity correction, used to compare the curves of ATAC-seq accessibility in GAF_control and GAF_degradFP from −500 bp to +500 bp from the center of the peaks. **c** Heatmaps of *k*-means clustered mitotically retained GAF peaks, based on H3K27ac and H3K27me3 ChIP-seq[56] and RNA-seq[57] from n.c. 14 embryos. *n*: number of identified GAF peaks. **d** Heatmaps representing the mitotic ATAC-seq signal[54] (dark blue) from interphase n.c.13 embryos, the ChIP-seq enrichment of H4K8ac in mitotic embryos and the ChIP-seq enrichment of Polycomb (Pc)[51] at the clustered mitotically retained GAF peaks from (**c**). *n*: number of identified GAF peaks.

available ATAC-seq data[54]. We observed that GAF mitotically retained regions are globally more open than GAF interphase only or mitotic-only targets (Fig. 3a).

More specifically, chromatin accessibility at mitotically retained regions encompasses larger regions than at loci bound by GAF only during interphase. This is in agreement with mitotically retained regions exhibiting a larger number of GAGA binding sites, potentially reflecting an enhanced number of bound GAF proteins able to foster nucleosome eviction (Supplementary Fig. 3a). Moreover, mitotically retained loci open gradually across developmental time windows and remain accessible during mitosis (Fig. 3a). Global chromatin accessibility at GAF mitotically retained targets is mostly linked to accessibility at *cis*-regulatory regions (Supplementary Fig. 3b).

We then asked whether chromatin accessibility at GAF mitotically retained regions required the presence of GAF. For this, we used ATAC-seq data performed on embryos where GAF levels were significantly reduced[28]. From this dataset, we retrieved GAF-bound loci for which accessibility was shown to be dependent on GAF. We found that the vast majority of these GAF-dependent regions (96%) correspond to GAF targets that we identified as mitotically retained (Supplementary Fig. 3d). Interestingly, targets depending on GAF for their accessibility mostly coincide with TSS and enhancer regions but do not overlap TAD boundaries[51] (Supplementary Fig. 3c). Importantly, interphase GAF targets or Zelda-only bound targets (not bound by GAF) did not show such a dependency on GAF for their accessibility (Fig. 3b). Collectively, these results suggest that GAF retention at

specific promoters and enhancers during mitosis may foster an accessible chromatin organization, which resists the overall compaction of the genome occurring during mitosis. However, other factors in addition to GAF are likely to foster chromatin accessibility during mitosis.

**GAF mitotic-bound regions are enriched with active and repressive histone marks**. GAF is known to be present in both active and repressive chromatin regions[33,55]. We, therefore, assessed the chromatin landscape of GAF mitotically retained regions. For this purpose, we focused on embryonic ChIP-seq profiles of characteristic chromatin marks: H3K27ac for active chromatin state and H3K27me3 for the repressed chromatin state[56], as well as RNA-seq signal from nc14 embryos[57]. By clustering GAF mitotically retained regions, we partitioned GAF targets into three distinct clusters (Fig. 3c and Supplementary Fig. 3e). The first cluster (44% of mitotically retained GAF) corresponds to GAF mitotic peaks with significant enrichment in H3K27ac, depleted in H3K27me3, and with a high RNA-seq signal. In contrast, the second cluster (26% of mitotically retained GAF peaks) displayed enrichment for H3K27me3 concomitant with depletion in H3K27ac and low RNA-seq signal. The remaining GAF mitotic targets fall into a third cluster (30% of mitotically retained GAF peaks), which displays no particular epigenetic features with our clustering analysis but shows significantly less chromatin accessibility (Fig. 3d). To examine if additional chromatin marks could discriminate between these three GAF clusters we performed ChIP-seq on the acetylation of lysine 8 of histone H4 (H4K8ac). Indeed among the myriad of chromatin marks labeling active regions, H4K8ac is a prominent mark during the initial reshaping of the genome during *Drosophila* ZGA[56]. We used our mitotic ChIP-seq method (Fig. 2a) to map H4K8ac in interphase and mitotic embryos genome-wide (Supplementary Fig. 4a–d). We observed that H4K8ac was particularly enriched in cluster 1 (Fig. 3d and Supplementary Fig. 4e).

Since cluster 2 was enriched for the Polycomb-associated mark H3K27me3, and as GAF is known to bind Polycomb Response Elements (PREs)[58], we asked if cluster 2 was enriched for PREs. Assessing the distribution of Polycomb (Pc) protein[59], a known Polycomb Group protein component specifically recruited at PRE in *Drosophila*[60], indeed confirmed that cluster 2 was highly enriched for Pc occupancy (Fig. 3d).

Together, these results demonstrate that mitotic GAF retention occurs at genomic regions associated with both active or repressive chromatin states. We propose that the combinatorial action of GAF and histone marks, contribute to the selective mitotic bookmarking of active regions to propagate transcriptional programs across cellular divisions.

**GAF mitotic bookmarking is not associated with mitotic loops**. Strictly speaking, mitotic occupancy by a TF can be envisaged as a mitotic bookmark only if it leads to a functional "advantage" upon mitotic exit. Because chromatin loops between *cis*-regulatory regions were observed to be re-established by late anaphase/telophase in mammalian cells[10] and since GAF is implicated in loop formation in *Drosophila*[61,62], we asked if GAF mitotically bound loci could form loops during mitosis in the embryo. We first focused on a specific genomic region containing two developmental genes, *charybde* (*chrb*) and *scylla* (*scyl*), separated by 235 kb and bound by GAF during both interphase and mitosis (Supplementary Fig. 5a, b). These early expressed genes were previously shown to form a long-range chromatin loop during early development[63].

We first confirmed that these loci are physically close and form a loop in nc14 by DNA FISH (Supplementary Fig. 5c). Interestingly, this proximity seems to be reinforced during nc14 progression (Supplementary Fig. 5c). However, while there is an overall genome compaction during mitosis, the distance between *scyl* and *chrb* is not different from that of a control locus, in mixed stages of mitosis (Supplementary Fig. 5c). To confirm this result, we examined two other loci using DNA FISH and assessed their potential looping across the cell cycle (Supplementary Fig. 5d). Both *snail* (*sna*) and *escargot* (*esg*) show GAF binding and the H4K8ac mark in interphase and mitosis. While these loci, form a loop in interphase nuclei; this long-range loop is not different from the control locus during mitosis (Supplementary Fig. 5e).

We, therefore, conclude that, at least for these regions, GAF mitotic binding is not associated with detectable stable mitotic DNA looping.

**The GAF bookmarked *scyl* gene harbors transcriptional memory**. To test if GAF fosters rapid postmitotic reactivation, we employed quantitative imaging on a selected GAF mitotically bound target, the zygotically expressed gene *scylla* (*scyl*). This gene is regulated by a promoter/proximal enhancer containing six GAGAG motifs, bound by GAF during interphase and mitosis (cluster 1 of mitotically retained loci) (Fig. 4a). To follow transcription dynamics with the high temporal resolution, we utilized the MS2/MCP signal amplification method[14,15] and quantitative imaging in living embryos. An array of 24X-MS2 repeats was inserted by CRISPR/Cas9 gene editing into the 3'UTR of *scyl* (Fig. 4a). MS2 reporter expression follows *scyl* endogenous expression (Fig. 4b and Supplementary Fig. 6a), and homozygous *scyl-MS2* stocks are viable and fertile. Then, we monitored postmitotic gene reactivation in nc14 in the ventral (Fig. 4c and Supplementary Fig. 6d) and dorsal side (Supplementary Fig. 6c). In both locations, postmitotic activation was found to be relatively fast, with a lag time of only 7.5 min and 9 min to reach 50% of the full pattern of activation (t50) in the dorsal ectoderm (Supplementary Movie 4) and mesoderm (Supplementary Movie 3), respectively (Supplementary Fig. 6d).

In addition to this temporal information within a given interphase, live imaging of transcription in the context the fast-developing *Drosophila* embryo gives access to nuclei genealogy. We assessed whether the transcriptional status of mother nuclei (prior to division) influences that of their descendants[64]. Indeed, we have previously shown that within the mesoderm, descendants of active nuclei in nc13 activate transcription significantly faster than those arising from inactive nuclei, a bias named "transcriptional memory"[18]. However, this was shown in the context of reporter transgenes and has thus far never been demonstrated at an endogenous locus.

To assess the existence of transcriptional memory at an endogenously mitotically bookmarked locus, we imaged *scyl* expression in the mesoderm. Within this domain, the expression was stochastic in nc13 (Fig. 4b and Supplementary Fig. 6c, and Supplementary Movie 3), allowing unambiguous discrimination between active and inactive mother nuclei prior to mitosis. By tracking the timing of activation for daughters arising from active mother nuclei compared to those coming from inactive mother nuclei (Fig. 4c), we observe a clear transcriptional memory bias (Fig. 4d and Supplementary Fig. 6e).

In order to test if this bias was due to a stronger activity of the *scyl* gene in nuclei coming from active mothers, we examined instantaneous intensities of transcriptional sites as they are directly correlated to the mRNA synthesis efficiency. Once active, instantaneous transcriptional site intensities were similar in nuclei coming from active mothers compared to those coming

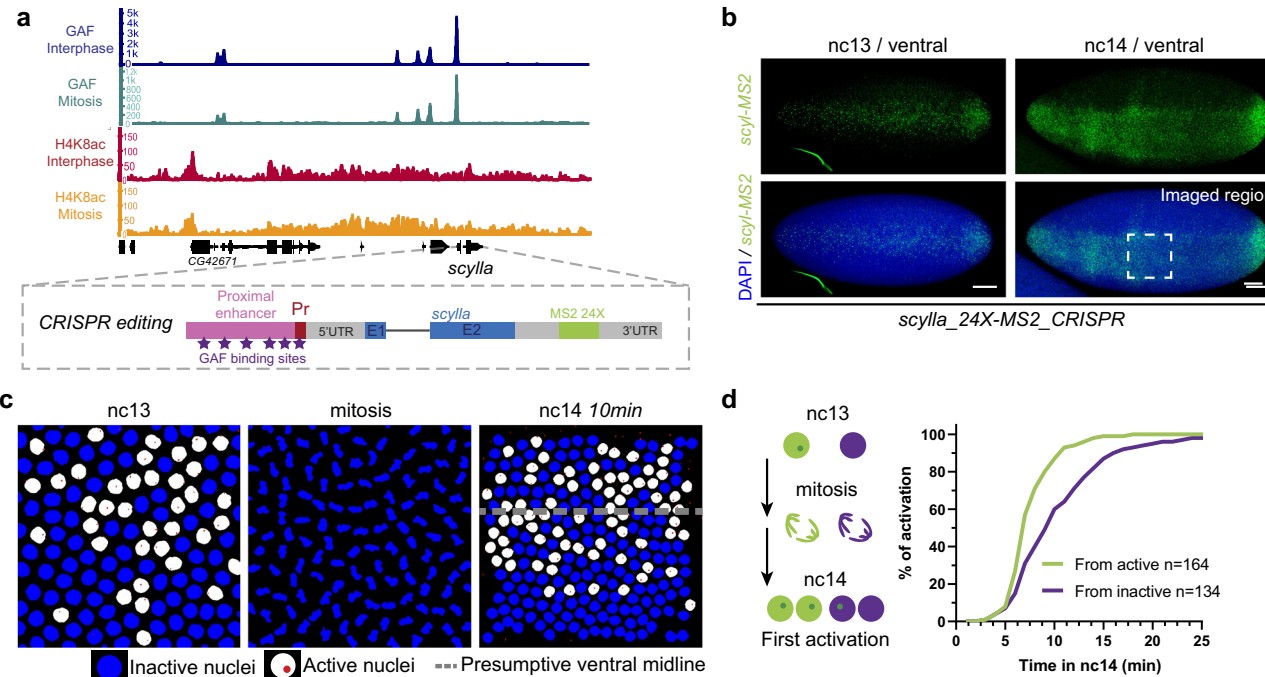

**Fig. 4 *scylla* gene harbors a transcriptional memory across mitosis. a** (Top) Genome browser image of interphase and mitotic GAF (dark blue and turquoise) and H4K8ac (red and orange) ChIP-seq signal at the *scyl* locus. (Bottom) Schematic of the 24X-MS2 tagging strategy of the *scyl* locus by CRISPR editing. A unique guide RNA sequence was designed to insert the 24X-MS2 repeats by homologous recombination, at 70 bp from the stop codon. **b** Maximum intensity projected Z-planes of confocal images from smiFISH with MS2 probes (green) counterstained with DAPI (blue) of *scylla_24X-MS2_CRISPR/+* embryos in nc13 and nc14. Scale bars are 50 μm. The dashed box represents the region considered for live imaging experiments. **c** Snapshots from a representative false-colored movie of *scylla_24X-MS2_CRISPR/+* embryo carrying *MCP-eGFP, His2Av-mRFP*. Active nuclei are represented in white and inactive nuclei in blue. Transcriptional sites are false-colored in red. The dashed line represents the presumptive ventral midline. **d** Quantification of transcriptional memory for *scyl* gene. Left panel: schematic of the two populations of nuclei studied; those derived from active (in green) and those from inactive nuclei (purple). Right panel: cumulative activation of the first activated nuclei coming from active nuclei (green) and from inactive (purple). n = number of analyzed pooled nuclei from four movies of four independent embryos.

from inactive mothers (Supplementary Fig. 6f). To describe the location of transcriptional activation of *scylla* with respect to GAF concentration, we performed immuno-RNA FISH with GAF antibody and MS2 probes on *scylla_24XMS2* embryos (Supplementary Fig. 6g). While GAF large puncta are located apically (see also Supplementary Fig. 1a, c), MS2 transcription foci are not overlapping and are located in the middle of the nuclear space. However, we cannot exclude that a subset of MS2 foci might colocalize with smaller GAF foci.

**GAF knockdown delays postmitotic transcriptional reactivation**. To test whether GAF was involved in the establishment of transcriptional memory, we employed RNAi knockdown (KD) to reduce the pool of maternal GAF. As previous studies reported difficulties to successfully deplete maternal GAF using a specific set of Gal4 driver[65], we decided to increase the efficiency of our depletion by combining two strong Gal4 drivers (*mat-alphaTub-Gal4* and *nanos-Gal4*). This strategy induces RNAi at all steps of oogenesis[66]. The level of maternal GAF mRNA KD was estimated to be 88% by qRT-PCR (Supplementary Fig. 7a) and also confirmed by western blot (Supplementary Fig. 7b), creating a substantial embryonic lethality. However, in this genetic context, a few embryos survived until gastrulation, albeit with clear mitotic and patterning defects for GAF targets genes (Supplementary Movie 6 and Supplementary Fig. 7c).

By quantifying postmitotic reactivation timing of *scyl* in *RNAi-GAF* embryos (Supplementary Movies 5 and 6), we observed a delay of ~6 min for t50 (Fig. 5a). We then compared the kinetics of activation in the two subpopulations (from active and from

inactive) and found that the transcriptional memory bias was reduced in *RNAi-GAF* embryos (Fig. 5b, c). Such a memory reduction does not occur upon maternal depletion of the pioneer factor Zelda[31], despite a slowdown of overall transcriptional dynamics.

Collectively, these data demonstrate that GAF controls the timing of transcriptional activation after mitosis and participates in the establishment of transcriptional memory.

**Modeling GAF driven transcriptional memory**. We analyzed the statistical distribution of the postmitotic delay (waiting times), defined as the lag time between the end of mitosis and the first activation in nc14 (Supplementary Fig. 8a). We have previously developed a simple mathematical model of memory, where this delay was modeled by a mixed gamma distribution[31] with two main parameters, the average number of rate-limiting transitions prior to reaching the transcription active state (ON) (parameter "a") and their durations (parameter "b"). Applying this mathematical model to our live imaging movies of *scyl* transcription dynamics in control (*RNAi-white*) and in GAF-depleted embryos (*RNAi-GAF*) (Supplementary Fig. 8b and Supplementary Data 4) revealed that the "a" parameter was comparable across genotypes (Supplementary Fig. 8c). However, upon GAF KD, the "b" parameter significantly increased in nuclei coming from active mother nuclei (Supplementary Fig. 8c). Remarkably, this selective decrease in the "b" parameter within a subpopulation was not observed upon Zelda depletion[31]. In order to be able to compare the effect of various genotypes, subject to distinct *cis*-regulatory codes, we introduced a memory score defined by the ratio $(ab_{inactive})/(ab_{active})$. A memory

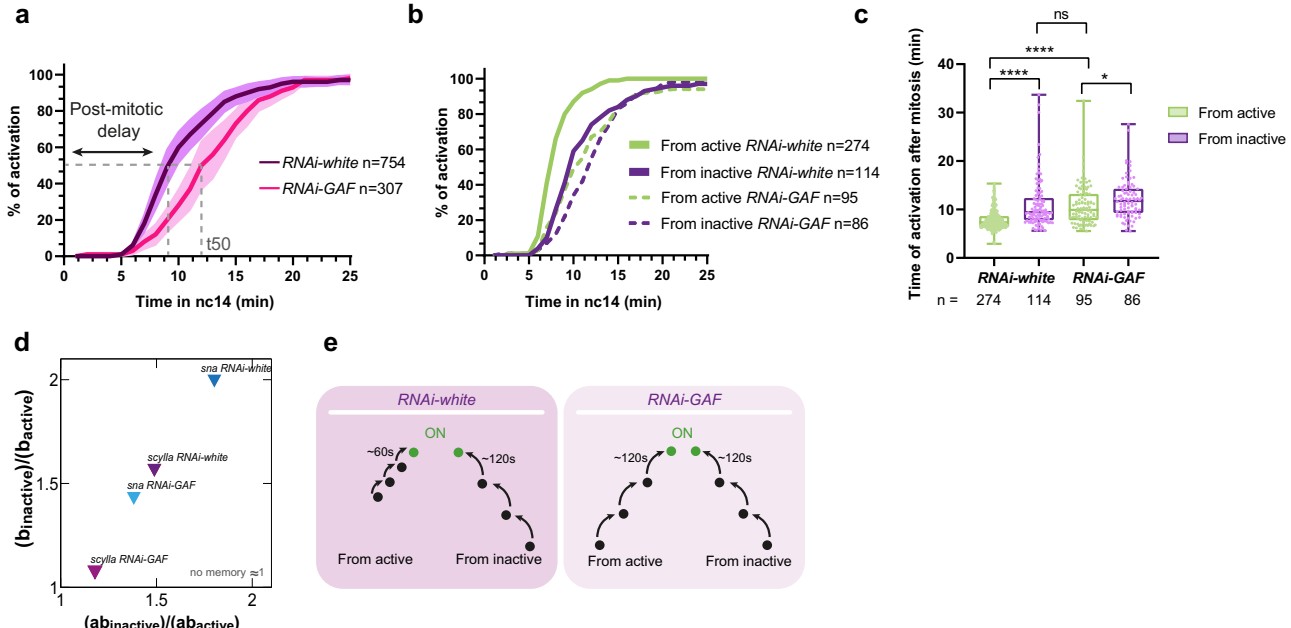

**Fig. 5 GAF is required for transcriptional memory of *Scylla*. a** Quantification of transcriptional synchrony of *scylla_24X-MS2_CRISPR/+* embryo after mitosis in *RNAi-white* (control, purple) and *mat-alphaTub-Gal4/+; nos-Gal4/UASp-shRNA-GAF* embryos (pink). The dashed line represents the t50 where 50% of the pattern is activated during nc14. Both of the two daughters derived from each nucleus are quantified. SEM are represented in light purple and light pink. *n* = number of nuclei analyzed from four movies of four independent embryos for each condition. **b** Cumulative activation of the first activated nuclei coming from active nuclei (green) and from inactive nuclei (purple) in *RNAi-white* embryos (control, solid curves) and *RNAi-GAF scylla_24X-MS2_CRISPR/+* embryos (dashed curves). *n* = number of pooled nuclei analyzed from four movies of four embryos. **c** Box plot representing the mean time of the first activation after mitosis of nuclei derived from active (green) and inactive (purple) nuclei in *RNAi-white* embryos and *RNAi-GAF scylla_24X-MS2_CRISPR/+* embryos. The centered horizontal line represents the median, whiskers represent min and max values. Two-tailed Welch's *t* test ****$P < 0.0001$, *$P = 0.0252$. **d** Ratios of parameter "b" and "ab" in subpopulations from inactive and active nuclei of *scylla_24X-MS2_CRISPR/+* (purples) and *snail-primary-enhancer_MS2* (blues) in *RNAi-white* or *RNAi-GAF* embryos. The parameter "a" corresponds to the average number of transitions (provided by the sum of weighted probabilities) and the parameter "b" to the time of each jump from one state to another. **e** Schematic of the proposed role of GAF in transcriptional memory. In the presence of GAF, nuclei derived from active nuclei have shorter "b" length than those derived from inactive nuclei whereas in the absence of GAF, both have the same transition times.

bias exists when this ratio is higher than 1. Using this metric, we observe that endogenous *scyl* exhibits a clear memory bias that vanishes upon GAF depletion (Fig. 5d). Interestingly, a GAF-dependent memory bias was also observed with a second GAF mitotically bound region (*sna-proximal-enhancer*, cluster 1, Fig. 2c', see "Methods") (Fig. 5d and Supplementary Fig. 8d). In all cases, we observe $(ab_{inactive})/(ab_{active}) \approx b_{inactive}/b_{active}$ (Fig. 5d), suggesting that the primary contribution to the memory bias comes from the transition duration "b".

Collectively, these results suggest a model where transcriptional memory bias results from distinct epigenetic paths in nuclei where a given locus is bookmarked by GAF and in nuclei where the same locus is not bound by GAF (Fig. 5e). The preferential bookmarking of active nuclei by GAF could be explained by stochastic GAF binding. We speculate that, during the interphase of nc13, there would be a differential probability of GAF binding between active and inactive mother nuclei. This differential in GAF binding in interphase of nc13 would persist during mitosis (our data suggest that GAF residence time is long) and would explain why descendants of active nuclei, can activate transcription faster than those coming from inactive (GAF-unbound during mitosis) nuclei.

## Discussion

We set out to determine how gene regulation by a transcription factor might be propagated through mitosis in a developing embryo. By using a combination of quantitative live imaging and genomics, we provide evidence that the pioneer-like factor GAF

acts as a stable mitotic bookmarker during zygotic genome activation in *Drosophila* embryos.

Our results indicate that during mitosis, GAF binds to an important fraction of its interphase targets, largely representing *cis*-regulatory sequences of key developmental genes (Supplementary Data 2 and 3). We noticed that GAF mitotically retained targets contain a larger number of GAGA repeats than GAF interphase-only targets and that this number of GAGA repeats correlates with the broadness of accessibility. Multiple experiments, with model genes in vitro (e.g., *hsp70*, *hsp26*) or from genome-wide approaches clearly demonstrated that GAF contributes to the generation of nucleosome-free regions[33]. The general view is that this capacity is permitted through the interaction of GAF with nucleosome remodeling factors as PBAP (SWI/SNIF), NURF (ISWI)[38], or FACT[67]. Although not yet confirmed with live imaging, immunostaining data suggest that NURF is removed during metaphase but re-engages chromatin by anaphase[68]. If the other partners of GAF implicated in chromatin remodeling are evicted during early mitosis, chromatin accessibility at GAF mitotic targets could be established prior to mitosis onset and then maintained through mitosis owing to the remarkable stability of GAF binding. However, we cannot exclude GAF interactions with other chromatin remodelers (e.g., PBAP) during mitosis and a scenario whereby mitotic accessibility at GAF targets would be dynamically established during mitosis thanks to the coordinated action of GAF and its partners.

We propose that the function of GAF as a mitotic bookmarker is possible because GAF has the intrinsic property to remain

bound to chromatin for long periods (residence time in the order of minute). This long engagement of GAF to DNA is in sharp contrast with the binding kinetics of many other TF, such as Zelda or Bicoid in *Drosophila* embryos[31,45] or pluripotency TF in mouse ES cells[7,69]. Another particularity of GAF binding, contrasting with other TF, resides in the multimerization of its DNA-binding sites as GAGAG repeats in a subset of its targets (76% of mitotically retained peaks display four or more repetitions of GAGAG motifs). Given the known oligomerization of GAF[70] and as GAF is able to regulate transcription in a cooperative manner[71], it is tempting to speculate that GAF cooperative binding on long stretches of GAGAG motifs may contribute to a long residence time.

Collectively, we propose that the combination of long residence time and the organization of GAF-binding sites in the genome may allow the stable bookmarking of a subset of GAF targets during mitosis.

In this study, we also discovered that a combination of GAF and histone modification could be at play to maintain the chromatin state during mitosis. Indeed, mitotic bookmarking may also be supported by the propagation of histone tail modifications from mother to daughter cells. Work from mammalian cultured cells revealed widespread mitotic bookmarking by epigenetic modifications, such as H3K27ac and H4K16ac[72,73]. Moreover, H4K16ac transmission from maternal germline to embryos has recently been established[74]. In the case of GAF, we propose that the combinatorial action of GAF and epigenetic marks, possibly selected via GAF interacting partners, will contribute to the propagation of various epigenetic programs. It would be therefore interesting to employ our established mitotic ChIP method to survey the extent to which *cis*-regulatory regions exhibit different mitotic histone mark modifications during embryogenesis.

A key aspect of mitotic bookmarking is to relate mitotic binding to the rapid transcriptional activation after mitosis. Here we show that GAF plays a role in the timing of reactivation after mitosis. However, we note that GAF binding during mitosis is not the only means to accelerate gene activation. Indeed, we and others have shown that mechanisms such as enhancer priming by Zelda, paused polymerase or redundant enhancers contribute to fast gene activation[75,76]. Moreover, a transcriptional memory bias can occur for a transgene not regulated by GAF[18]. By modeling the transcriptional activation of the gene *scylla*, we reveal that GAF accelerates the epigenetic steps prior to activation, selectively in the descendants of active nuclei. We propose a model where GAF binding helps in the decision-making of the postmitotic epigenetic path. In this model, mitotic bookmarking by GAF would favor an epigenetic path with fast transitions after mitosis (Fig. 5e). In the context of embryogenesis, bookmarking would lead to the fast transmission of select epigenetic states and may contribute to gene expression precision.

Interestingly, GAF vertebrate homolog (vGAF/Th-POK) has recently been implicated in the maintenance of chromatin domains during zebrafish development[77]. We, therefore, suspect that GAF action as a stable bookmarking factor controlling transcriptional memory during *Drosophila* ZGA might be conserved in vertebrates.

## Methods

**Fly handling and genetics**. The *yw* stock was used as a wild-type. The germline driver *nos-Gal4:VP16*(BL4937) was previously recombined with a *MCP-eGFP-His2Av-mRFP* fly line[31]. RNAi were expressed after crossing this recombinant for live imaging (or *nos-Gal4:VP16* for fixed experiments) with Gal4 under the expression of *maternal-alphaTubulin* promoter (*mat-alphaTub-Gal4* (BL7063)), then with *UASp-shRNA-w* (BL35573) or *UASp-shRNA-GAF* (BL41582). Virgin females expressing RNAi, MCP-GFP-His2Av-mRFP and both Gal4 constructs were crossed with MS2 containing CRISPR alleles or transgene-containing males. All experiments were done at 21 °C except RNAi experiments which were done at

25 °C. The C-terminal tagged version of GAF-sfGFP was obtained by CRISPR/Cas9[28].

**Cloning and transgenesis**. The *snail-primary-enhancer_MS2* transgene was obtained by amplification of the *sna* endogenous promoter and primary enhancer using the primers listed in Supplementary Data 1. The 128XMS2 tag[78] was inserted immediately upstream of the yellow reporter gene sequence of the pbphi-yellow plasmid[17]. The transgenic construct was inserted in the VK0033 landing site (BL9750) using PhiC31 targeted insertion[79].

The homology arms for the recombination template for CRISPR/Cas9 editing of *scyl* gene to generate *scyl_24X-MS2_CRISPR* were assembled with NEBuilder® HiFi DNA Assembly Master Mix (primers listed in Supplementary Data 1) and inserted into pBluescript opened *SpeI/AscI* (for the 5′ homology arm) or *XmaI/NheI* (for the 3′ homology arm) containing the 24X-MS2 (as in ref. [31]) inserted after *Not1* digestion. Guide RNA (Supplementary Data 1) were cloned into pCFD3-dU6:3gRNA (Addgene 49410) digested by BbsI using annealed oligonucleotides (Integrated DNA Technology™). The recombination template and guide RNA plasmids were injected into BDSC#55821 (BestGene Inc.). Transformant flies were screened using a dsRed marker inserted downstream of the 3′UTR of the genes.

**Fluorescence recovery after photobleaching**. Fluorescence recovery after photobleaching (FRAP) in embryos at nc14 was performed on a Zeiss LSM880 using a 40 × /1.3 Oil objective and a pinhole of 84 μm. Images (256 × 128 pixels, 16bits/pixel, zoom ×6) were acquired every ≈53 ms for 1200 frames. GFP was excited with an Argon laser at 488 nm and detected between 492 and 534 nm. The laser power of the 488 nm laser for FRAP acquisition images was 5 μW. Measurements are taken with a ×10 objective. Laser intensity was kept as low as possible to minimize unintentional photobleaching. A circular ROI (12 × 12 pixels) 0.138 μm/pixel, was bleached using two laser pulses at maximal power during a total of ≈110 ms after ten frames. To discard any source of fluorescence intensity fluctuation other than molecular diffusion, the measured fluorescence recovery in the bleached ROI region ($I_{bl}$) was corrected by an unbleached ROI ($I_{unbl}$) of a neighbor's nucleus and another ROI outside of the nucleus ($I_{out}$) following the simple equation:

$$I_{bl_{corr}}(t) = \frac{I_{bl}(t) - I_{out}(t)}{I_{unbl}(t) - I_{out}(t)} \quad (1)$$

The obtained fluorescence recovery was then normalized to the mean value of fluorescence before the bleaching i.e.,

$$I_{bl_{norm}}(t) = \frac{I_{bl_{corr}}(t)}{\frac{1}{N}\sum_{n=1}^{10} I_{bl}(n)} \quad (2)$$

Analytical equations used to fit the fluorescence recovery was chosen with two exchanging population on the first 1100 frames: we started from the analytical expression developed in Supplementary Eq. (35) of ref. [80].

$$F(t) = F_{eq}F_D(t) + C_{eq}F_{exc}(t) \quad (3)$$

with $C_{eq}$ defined as above and $F_{eq} = k_{off}/(k_{off} + k^*_{on})$. $F_D(t)$ is the fluorescence recovery due to diffusion and $F_{exc}(t)$ the fluorescence recovery due to exchange.

Since we used a Gaussian shape illumination profile, $F_D(t)$ is defined using a slightly modified version of the analytical equation of the 20th order limited development of the Axelrod model for Gaussian profile illumination and diffusion[81,82]:

$$F_D(t) = \frac{1 - e^{-K}}{K}(1-M) + M\sum_{n=1}^{20} \frac{(-K)^n}{n!}\left(1 + n + 2n\frac{t}{\tau}\right)^{-1} \quad (4)$$

$$M = \frac{I_{(t>30\tau)} - I_0}{1 - I_0} \quad (5)$$

where $K$ is a constant proportional to bleaching deepness, $M$ is the mobile fraction and $\tau$ is the half time of recovery. To minimize the effect of the mobile fraction on $C_{eq}$, $M$ was kept between 0.9 and 1.1.

Diffusion coefficients of the different molecules were determined according to

$$D = \frac{\beta w^2}{4\tau} \quad (6)$$

with w the value of the radius at $1/e^2$ of the Gaussian beam (in our case, $w = 0.83$ μm) and β a discrete function of $K$ tabulated in ref. [83].

$F_{exc}(t)$ is defined as in ref. [80], slightly modified with respect to the Gaussian illumination, leading to the following equation:

$$F_{exc}(t) = F_\infty - \left(\frac{1 - e^{-K}}{K} - F_\infty\right)e^{-k_{off}t} \quad (7)$$

with $K$ defined as previously.

FRAP curve fitting was done with MatLab 2014b (Mathworks Inc. USA).

**Fluorescence correlation spectroscopy**. Fluorescence correlation spectroscopy (FCS) experiments were performed on a Zeiss LSM780 microscope using a ×40/1.2 water objective. GFP was excited using the 488 nm line of an Argon laser with a pinhole of 1 airy unit. Intensity fluctuation measured for 10 s were acquired and

autocorrelation functions (ACFs) generated by Zen software were loaded in the PyCorrFit program[84]. Multiple measurements per nucleus in multiple nuclei and embryos at 20 °C were used to generate multiple ACF, used to extract parameters. The FCS measurement volume was calibrated with a Rhodamine6G solution[85] using $D_f = 414 \, \mu m^2 \, s^{-1}$. Each time series was fitted with the following generic equation:

$$G(\tau) = 1 + \frac{1}{N} \left( 1 + \frac{Te^{-\frac{t}{\tau_T}}}{1-T} \right) \left( \sum_{i=1}^{n} \frac{f_i}{\left(1 + \frac{t}{\tau_i}\right)\left(1 + \frac{t}{s^2 \tau_i}\right)^{1/2}} \right) + G \qquad (8)$$

Using $n = 2$ in our fit and where $N$ is the total number of molecules, $T$ is the proportion of the fluorescent molecules $N$ in the triplet state with a triplet state lifetime $\tau_T$ (constrained below 10 μs in our fit), $f_i$ is the proportion of each different diffusing species ($\sum_{i=1}^{n} f_i = 1$) with a diffusion time $\tau_i = w^2_{xy} / 4 D$ and $s^2 = w_z / w_{xy}$. We also introduced a $G_\infty$ value to account for a long time persistent correlation during the measurements.

**Immunostaining.** A pool of 0–4 h after egg-laying (AEL) or 2–4 h AEL embryos were dechorionated with bleach for 3 min and thoroughly rinsed with $H_2O$. They were fixed in 1:1 heptane:formaldehyde-10% for 25 min on a shaker at 450 rpm; formaldehyde was replaced by methanol and embryos were shaken by hand for 1 min. Embryos that sank to the bottom of the tube were rinsed three times with methanol. For immunostaining, embryos were rinsed with methanol two times and washed three times 3 min with PBT (PBS 1 × 0.1% triton). Embryos were incubated on a wheel at room temperature for 30 min in PBT, then for 20 min in PBT 1% BSA, and at 4 °C overnight in PBT 1% BSA with primary antibodies. Embryos were rinsed three times, washed twice 20 min in PBT, then incubated in PBT 1% BSA for 20 min, and in PBT 1% BSA with secondary antibodies for 2 h at room temperature. Embryos were rinsed three times then washed three times in PBT for 10 min. DNA staining was performed using DAPI at 0.5 μg/ml. Primary antibody dilutions for immunostaining were mouse anti-GFP (Roche IgG1κclones 7.1 and 13.1) 1:200; mouse anti-H3K9me2-3 (gift from Dr. J. Dejardin) 1:300; rabbit anti-GAF (gift from Dr. G.Cavalli) 1:250; 1:100. Secondary antibodies (anti-rabbit Alexa 488-conjugated (Life Technologies, A21206); anti-mouse Alexa 488-conjugated (Life Technologies, A21202); anti-rabbit Alexa 555-conjugated (Life Technologies, A31572)) were used at a dilution 1:500. Mounting was performed in Prolong® Gold.

Images in Supplementary Fig. 1a represent a maximum intensity projection of a stack of 3 z-planes (≈1 μm). Images in Supplementary Fig. 1b represent a single Z-plane. Images in Fig. 1a represent a maximum intensity projection of a stack of 9 z-planes (≈4 μm).

**Single-molecule fluorescence in situ hybridization (smFISH) and immuno-smFISH.** Embryos were fixed as in the previous section, then washed 5 min in 1:1 methanol:ethanol, rinsed twice with ethanol 100%, washed 5 min twice in ethanol 100%, rinsed twice in methanol, washed 5 min once in methanol, rinsed twice in PBT-RNasin (PBS 1×, 0.1% tween, RNasin® Ribonuclease Inhibitors). Next, embryos were washed 4 times for 15 min in PBT-RNasin supplemented with 0.5% ultrapure BSA and then once 20 min in Wash Buffer (10% 20× SCC, 10% Formamide). They were then incubated overnight at 37 °C in hybridization buffer (10% formamide, 10% 20x SSC, 400 μg/ml E. coli tRNA (New England Biolabs), 5% dextran sulfate, 1% vanadyl ribonucleoside complex (VRC) and smFISH Stellaris probes against sna coupled to Quasar670 and/or FLAP-Y probes and/or GAF primary antibody. FLAP-Y probes against 24X-MS2 and scyl were prepared by duplexing 40 pmol of target-specific probes with 100 pmol FLAP-Y-Cy3 (or FLAP-Y-alexa488 for double 24X-MS2 and scyl FISH) oligonucleotides and 1× NEBuffer™ 3 for 3 min at 85 °C, 3 min at 65 °C and 5 min at 25 °C and kept on ice until use. Probe sequences are listed in Supplementary Data 1.

Embryos were washed three times 15 min in Wash Buffer at 37 °C (with the third wash done with DAPI). An extra wash with secondary antibody (1/500 anti-rabbit Alexa 488-conjugated (Life Technologies, A21206)) was added if necessary, for 45 min at 37° in Wash Buffer. Embryos were then washed in 2× SCC, 0.1% Tween at room temperature before being mounted in ProLong® Gold antifade reagent. Images were acquired using a Zeiss LSM880 confocal microscope with an Airyscan detector in SR mode with a 40× Plan-Apochromat (1.3 NA) oil objective lens or a 20x Plan-Apochromat (0.8NA) air objective lens. Images were taken with 1024 × 1024 pixels and Z-planes 0.5μm apart. GFP was excited using a 488 nm laser, Cy3 was excited using a 561 nm laser, Quasar670 was excited using a 633 nm laser.

Images in Fig. 4b and Supplementary Fig. 6a and c represent a maximum intensity projection of a stack of 15 z-planes (≈9.5 μm). Images shown in Supplementary Fig. 6g correspond to a single plane. Images in Supplementary Fig.1d represent a maximum intensity projection of a stack of 5 z-planes 0.3 μm apart (≈1.5 μm).

**H3S10ph immunostaining and mitotic embryos sorting.** A pool of 1h30-2h30 AEL embryos was fixed as for immunostaining except the fixation was in 1:1 heptane:1.8% formaldehyde/1X PBS (Thermo Scientific 28906) for exactly 10 min shaking at 450 rpm. Then embryos were rapidly quenched with 125 mM glycine

PBS-1x and shaken for 1 min by hand. An anti-phospho-Histone H3 (Ser10) antibody (Cell Signalling #9701) was used at a dilution 1:200. Anti-mouse Alexa 488-conjugated (Life technologies, A21202) was used as a secondary antibody at a dilution 1:500. Embryos were kept in PBT until sorting.

Sorting was done using a COPAS SelectInstrument (Biometrica) with the following parameters: sorting limit low: 1, high: 256; PMT control: Green 650, Yellow 425, and Red 800. A restricted area of sorting (with the highest green signal) was selected representing ≈8% of the total population. A container was placed at the output of the non-selected embryos in order to re-pass them through the sorter to collect non-green embryos corresponding to interphase embryos. Right after the sorting, embryos were manually checked under a Leica Z16 APO macroscope by placing them on a glass cup and using Drummond Microcaps® micropipettes to remove mis-sorted embryos individually. In all, 1000 embryos per tube were then dried by removing the PBT and kept at −80 °C.

**Chromatin immunoprecipitation and library preparation.** In total, 1000 embryos were homogenized in 1 ml of Buffer A (60 mM KCl, 15 mM NaCl, 4 mM MgCl₂, 15 mM HEPES (pH 7.6), 0.5% Triton X-100, 0.5 mM DTT, 10 mM Sodium Butyrate and Protease Inhibitors Roche 04693124001) using a 2 ml Dounce on ice. The solution was then centrifuged 4 min at 2000 × g at 4 °C. The supernatant was removed and 1 ml of Buffer A was added and this was repeated two times with Buffer A and once with Lysis Buffer without SDS (140 mM NaCl, 15 mM HEPES (pH 7.6), 1 mM EDTA (pH 8), 0.5 mM EGTA, 1% Triton X-100, 0.5 mM DTT, 0.1% sodium deoxycholate, 10 mM sodium butyrate and protease inhibitors). The pellet was resuspended in 200 μl of Lysis Buffer with 0.1% SDS and 0.5% N-Laurosylsarcosine and incubated 30 min at 4 °C on a rotating wheel. Sonication was done with a Bioruptor® Pico sonication device with 30 sec ON/30 s OFF cycles for 6–7 min for interphase and 8–9 min for mitotic chromatin. Sonicated chromatin was then centrifuged 5 min at 14000 rpm at 4 °C. The chromatin was then diluted in 1 ml of Lysis Buffer.

Dynabeads® M-270 Epoxy (Invitrogen Life Technologies™, 14301) were prepared in order to directly cross-link antibodies to the beads (anti-GAF, gift from G. Cavalli, or anti-H4K8ac, Abcam 15823), avoiding cross-reaction with the H3S10ph antibody, following manufacturer protocol. Prior to this, anti-GAF was purified using NAb™ Protein A/G Spin Kit (Thermo Scientific). Once the magnetic beads were cross-linked, chromatin was incubated overnight at 4 °C on a rotating wheel. Then, beads were washed 7 min at 4 °C once in Lysis Buffer, once in FAT Buffer (1 M Tris-HCl pH 8, 0.5 M EDTA pH 8, SDS 10%, 5 M NaCl, 10% Triton), once in FA Buffer (1 M HEPES, pH 7.0–7.6, 5 M NaCl, 0.5 M EDTA pH 8, Triton X-100—10% NaDeoxycholate) once in LiCL Buffer (1 M Tris-HCl pH 8, 4 M LiCl, 10% Nonidet-P40-Nonidet, 10% NaDeoxycholate and protease inhibitors) and twice in TE (10 mM Tris-HCl pH 8, 0.1 mM EDTA). Elution was done in elution Buffer 1 (10 mM EDTA, 1% SDS, 50 mM Tris-HCl pH 8) for 30 min at 65 °C at 1300 rpm. Eluted chromatin was removed and a second elution step with Elution Buffer 2 (TE, 0.67% SDS) was performed. The two elutions were pooled. Chromatin was then reverse-cross-linked by heating overnight at 65 °C. Next, chromatin was incubated 3 h at 50 °C with ProteinaseK (Thermo Scientific™ EO0491) and RNAseA (Thermo Scientific™ EN0531). DNA was then extracted with phenol/chloroform purification. Biological duplicates were performed for each sample.

Libraries were then prepared using the NEBNext UltraII DNA Library Prep Kit for Illumina, following the manufacturer's instructions. Sequencing was performed on Illumina HiSeq 4000 on pair-end 75 bp.

**ChIP-seq analysis.** Both reads from ChIP-seq and Input experiments were trimmed for quality using a threshold of 20 and filtered for adapters using Cutadapt (v1.16). Reads shorter than 30 bp after trimming were removed. Reads were mapped to *Drosophila melanogaster* genome (dm6 release) using Bowtie2[86]. Aligned sequences were processed with the R package PASHA to generate the used wiggle files[87]. Pasha elongates in silico the aligned reads using the DNA fragment size estimated from paired reads. Then, the resulting elongated reads were used to calculate the coverage score at each nucleotide in the genome. Wiggle files representing the average enrichment score every 50 bp were generated. In order to normalize the enrichment scores to reads per million, we rescaled the wiggle files using PASHA package. Besides, in order to reduce the overenrichment of some genomic regions due to biased sonication and DNA sequencing, we subtracted from ChIP sample wiggle files the signal present in Input sample wiggle files. The Rescaled and Input subtracted wiggle files from biological replicate were then used to generate the final wiggle file representing the mean signal.

In order to call the enriched peaks from the final wiggle files, we used *Thresholding* function of the Integrated Genome Browser (IGB) to define the signal value over which we consider a genomic region to be enriched compared to background noise (*Threshold*). We used also the minimum number of consecutive enriched bins to be considered an enriched region (*Min.Run*) as well as the minimum gap above which two enriched regions were considered to be distinct (*Max.Gap*). The three parameters were then used with an in-house script that realizes peak calling by using the algorithm employed by *Thresholding* function of IGB.

Peaks calling was done with a threshold of 100 for GAF-ChIP-seq and 22 for H4K8ac-ChIP-seq, a minimum run of 50 bp and maximum gap of 200 bp.

Interphase-only peaks correspond to peaks from interphase ChIP-seq with no overlap with peaks from mitotic ChIP-seq. Mitotically retained correspond to interphase peaks with an overlap (min 1 base pair) with peaks from mitotic ChIP-seq. Mitotic-only peaks correspond to peaks from mitotic ChIP-seq with no overlap with peaks from interphase ChIP-seq.

Motif search was done with the MEME ChIP tool (MEME suite 5.1.1).

Peaks were considered as promoters if overlap with the region defined by 100 bp around TSS. Peaks were considered as enhancers if overlapping with identified enhancer[53] and/or overlapping with a H3K27ac peak[59].

ATAC-seq data are from ref. [54] (GSE83851). Wig files were converted to BigWig using Wig/BedGraph-to-bigWig converter (Galaxy Version 1.1.1). ATAC-seq mean signal was then plotted on regions of interest (mitotically retained peak coordinates and Interphase-only coordinates) using computeMatrix by centering ATAC-seq signal to the center of the regions (and $+/- 1$ kb) followed by plotProfile (Galaxy Version 3.3.2.0.0).

All studied features for the mitotically retained GAF peaks are summarized in Supplementary Data 3.

Mitotically retained GAF peaks were subdivided by $k$-means clustering based on chromatin state (H3K27ac and H3K27me3 ChIP-seq[56]) and transcriptional status (nc14 RNA-seq[57]) using deepTools[88]. Peaks were partitioned into three clusters: cluster 1, $n = 1073$, cluster 2, $n = 612$ and cluster 3, $n = 732$. To further characterize mitotically retained clusters we plotted heatmaps using deepTools[88] for publicly available ChIP-seq data for H3K27ac[56], H3K27me3[56], Pc[59], and ATAC-seq[54].

GAF-bound loci for which accessibility is dependent on GAF were taken from ref. [28]. All whole-genome data and stages are listed in Supplementary Table 1.

All ChIP-seq data and bed files are accessible at GEO Accession viewer under the number: GSE180812.

**Live imaging.** Movies of *His2Av-mRFP; sfGFP-GAF_CRISPR* (related to Supplementary Movies 1 and 2 and Fig. 1b) were acquired using a Zeiss LSM880 with a confocal microscope in fast Airyscan mode with a Plan-Apochromat ×40/1.3 oil objective lens. GFP and mRFP were excited using a 488 nm and 561 nm laser, respectively, with the following settings: 256 × 256-pixel images, 15 z-planes 1 μm apart, and zoom ×4, resulting in a time resolution of 9.5 s per z-stack. Average intensity profiles were measured for histones, nucleoplasmic GAF, and cytoplasmic GAF from three movies of embryos transitioning from nc13 into nc14. Maximum intensity projected images were used for automatic tracking using a homemade software as in ref. [31]. First, detection of nuclei is made using His2Av-mRFP allowing the monitoring of histone intensity fluctuation, then a mask of His2Av-mRFP detected nuclei was projected on the sfGFP-GAF channel allowing the recovery of sfGFP-GAF present on histones. Finally, five ROI in each movie corresponding to cytoplasmic regions were tracked for sfGFP-GAF intensity in the cytoplasm. The laser power is 12 μW for 488 nm laser and 22 μW for 561 nm laser. Measurements are taken with a ×10 objective.

Movies of *MCP-eGFP-His2Av-mRFP>snail-primary-enhancer_MS2/+* embryos (related to Supplementary Fig. 7e) were acquired using a Zeiss LSM780 with a confocal microscope with a Plan-Apochromat ×40/1.3 oil objective lens. GFP and mRFP were excited using a 488 nm and 561 nm laser respectively with the following settings: 512 × 512-pixel images, 21 z-planes 0.5 μm apart and zoom ×2.1, resulting in a time resolution of 22 s per frame. Movies subjected to filtering steps to track transcription foci as 128XMS2 loops result in signal retention during mitosis. The laser power is 4.5 μW for 488 nm laser and 10 μW for 561 nm laser. Measurements are taken with a ×10 objective.

Movies of *MCP-eGFP-His2Av-mRFP>scyl_MS2_CRISPR/+* in *wild-type* (Supplementary Movies 3 and 4) *RNAi-White* (Supplementary Movie 5) and *RNAi-GAF* (Supplementary Movie 6) background (related to Figs. 4 and 5 and Supplementary Figs. 6c–f and 7c) were acquired using a Zeiss LSM880 with a confocal microscope in fast Airyscan mode with a Plan-Apochromat ×40/1.3 oil objective lens. GFP and mRFP were excited using a 488 nm and 561 nm laser respectively with the following settings: 552 × 552-pixel images, 21 z-planes 0.5 μm apart and zoom ×2.1, resulting in a time resolution of 5.45 s per frame. As we observed that GAF knockdown was not complete (some *RNAi-GAF* embryos gastrulate and develop), movies showing visible developmental defects, such as nuclear dropout, anaphase bridges or failure to gastrulate, were kept for analysis. The laser power is 5 μW for the 488 nm laser and 14 μW for 561 nm laser. Measurements are taken with a ×10 objective.

**Memory movies analysis.** Movies were analyzed using Mitotrack[64] as in ref. [31] with newly implemented tools to filter mitotic 128XMS2 foci in movies of *MCP-eGFP-His2Av-mRFP>snail-primary-enhancer_MS2/+* embryos (mitotic foci are now detected with the 24MS2 array). Briefly, using a custom-made algorithm developed in Python™ and implemented in the MitoTrack software, nuclei were segmented and tracked in 2D, working on the maximum intensity projected stack. In order to detect spots in 3D, we perform a 3D Laplacian of Gaussian filter on the raw data, frame by frame, and threshold the filtered 3D images with a user-defined value. The threshold value is expressed as $\mu + thr * sigma$, where $\mu$ is the average and sigma is the standard deviation of the filtered images: thr is the user-defined value and it is common for all the analyses. The thresholded images are then masked with raw data, allowing to retrieve single nuclei intensities. All spots

present during mitosis were removed in the successive cycle such that only de novo appearing MS2 punctae were analyzed.

For intensity analysis (related to Supplementary Figs. 6f and 7c) the intensity of detected spots was collected for each frame to study the transcriptional intensity behavior throughout nuclear cycle 14. Transcription site intensity is given by the sum of the intensities in all the selected pixels[89]. Nuclei coming from inactive and nuclei coming from active were separated for Supplementary Fig. 6f. The background was measured in each movie using FIJI[90], by taking six different areas outside a transcriptional spot and calculating their mean. Next, intensity values were divided by the mean background of each movie.

The updated version of MitoTrack (with intensities measurements) is available at: https://github.com/ant-trullo/MitoTrack_v4_0.

**qRT-PCR in *RNAi* embryos.** Total RNA from 0 to 2 h AEL *RNAi-white* or *RNAi-GAF* driven by *nos-Gal4* and *mat-alphaTub-Gal4* embryos was extracted with TRIzol following the manufacturer's instructions. RNA was DNase-treated. In all, 1 μg of RNA extracted from ~300 embryos per replicate was reverse transcribed using SuperScript IV and random primers. Quantitative PCR analyses were performed with the LightCycler480 SYBR Green I Master system (primers used listed in Supplementary Data 1, targeting both isoforms of *GAF*). RNA levels were calculated using the *RpL32* housekeeping gene as a reference and not bound by GAF according to the GAF-ChIP-seq. Each experiment was performed with biological triplicates and technical triplicates.

**Western blot analysis.** Fifty embryos from *RNAi-white* or *RNAi-GAF* driven by *nos-Gal4* and *mat-alphaTub-Gal4* 0–2 h AEL embryos were collected and crushed in 100 μl of NuPAGE™ LDS sample buffer and reducing agent. Samples were heated 10 min at 70 °C, and the volume-equivalent of five embryos was loaded per well on a 4–12% Bis-Tris NuPAGE™ Novex™ gel and ran at 180 V. Protein transfer was done for 1 h10 min at 110 V to a nitrocellulose membrane, 0.2 μm (Invitrogen, LC2000). The membrane was blocked in 5% milk-PBT (PBS 1×0.1% Tween 20) for 40 min and incubated overnight at 4 °C with primary antibodies 1/2000 mouse anti-GAF or 1/2000 mouse anti-Tubulin (Invitrogen, GT114) in PBT. Anti-mouse and -rabbit IgG-HRP (Cell Signaling #7076 and #7074) secondary antibodies were used at 1/4000 and incubated 1 h at room temperature. Chemiluminescent detection was done using Pierce™ ECL Plus (ThermoFisher) kit.

Relative quantification in Supplementary Fig. 7b was performed on biological triplicates of *RNAi-white* and *RNAi-GAF* embryos. An area of 2508 pixels was taken around each band and the mean intensity signal was measured using Fiji[90]. Each value of GAF protein mean intensity signal was divided by the value of the Tubulin protein signal from the same sample.

**DNA probe preparation and DNA FISH.** Probes were generated using 4 to 6 consecutive PCR fragments of 1.2–1.5 kb from *Drosophila* genomic DNA, covering approximately a 10-kb region. Primers are listed in Supplementary Data 1. Probes were labeled using the FISH Tag DNA Kit (Invitrogen Life Technologies, F32951) with Alexa Fluor 488, 555, and 647 dyes following the manufacturer's protocol. Probes for satellite regions (related to Supplementary Fig. 1b) are from ref. [91].

DNA FISH was performed on 0–4 h AEL *yw* embryos adapted from[92]. Briefly, embryos were fixed as described above and were rehydrated with successive 3–5 min 1 ml washes on a rotating wheel with the following solutions: (1) 90% MeOH, 10% PBT; (2) 70% MeOH, 30% PBT; (3) 50% MeOH, 50% PBT; (4) 30% MeOH, 70% PBT; (5) 100% PBT. Embryos were subsequently incubated in 200 μg RNaseA (Thermo Scientific, EN0531) in 1 ml PBT for 2 h then 1 h at room temperature on a rotating wheel. Embryos were then slowly transferred to 100% pHM buffer (50% Formamide, 4× SSC, 100 mM NaH₂PO₄, pH 7.0, 0.1% Tween 20) in rotating wheel 20 min per solution 1 (1) 20% pHM, 80% PBS-Triton; (2) 50% pHM, 50% PBS-Triton; 80% pHM, 20% PBS-Triton; 100% pHM. Cellular DNA and probes were respectively denaturized in pHM and FHB (50% Formamide, 10% Dextran sulfate, 2× SSC, 0.05% Salmon Sperm DNA) for 15 min at 80 °C.

Probes and embryos were quickly pooled in the same PCR tube and slowly hybridized together with the temperature decreasing 1 °C every 10 min to reach 37 °C in a thermocycler. For peri-centromeric labeling, probes from[91] were used. Washes were performed in pre-warmed solution (1 to 4) at 37 °C for 20 min under 900 rpm agitation (1) 50% formamide, 10% CHAPS 3%, 10% SSC; (2) 40% Formamide, 10% CHAPS 3%, 10% SSC; (3) 30% formamide; (4) 20% formamide; then 20 min on a rotating wheel at room temperature using (5) 10% Formamide; (6) PBT; (7) PBS-Triton. For DNA-immunoFISH (with GAF antibody), embryos were proceeded as immunostaining protocol from here. Embryos were stained with DAPI at 0.5 μg/ml, washed in PBT, and mounted between slide and coverslip.

Images were acquired using a Zeiss LSM880 with a confocal microscope in Airyscan mode with a Plan-Apochromat ×63/1.4 oil objective lens with the following settings: zoom ×3.0, z-planes 0.3 μm apart, 1024 × 1024 pixels.

**Distance measurements.** To measure the distances between probes (*scyl-chrb* and *chrb-ctrl*, or *esg-sna* and *sna-ctrl*), we used custom-made software developed in Python™. This software is available through this link: https://github.com/ant-trullo/DNA_FishAnalyzer.

All probes channels were treated with a 3D Laplacian of Gaussian filter (with kernel size 1) and then detected in 3D with manual thresholding on the filtered matrices; for each of the detected spots, the center of mass was determined. DAPI signal was treated with a 3D Gaussian filter (with user-defined kernel size) and the logarithm of the resulting matrix is thresholded with an Otsu algorithm, the threshold value being adjusted separately in each frame. The logarithm was used in order to compensate for nonhomogeneous intensity inside nuclei. In order to generate distances, all the spots outside nuclei were removed. Then, nearest mutual neighbor spots were selected by calculating the distances of all the possible couples of spots and picking the smallest set. The distances were calculated with respect to the center of mass and using the Euclidean distance, taking into account the different pixel size on the $z$ axis. A minimum of ten images from five different embryos were analyzed for each condition. Aberrant distances (superior to 1 μm) were not considered.

**Mathematical modeling of mitotic memory**. We are interested in the postmitotic delay, defined as the time needed for postmitotic transcription (re)activation. We model this time as the sum of two variables as in ref. [31]:

$$T_a = T_0 + T_r \qquad (9)$$

where $T_0$ is a deterministic incompressible lag time, the same for all nuclei, and $T_r$ is a random variable whose value fluctuates from one nucleus to another. The decomposition in Eq. (9) can be justified by the experimental observation that all the reactivation curves (Fig. 5a and Supplementary Fig. 7c) start with a nonzero length interval during which no nuclei are activated. Furthermore, $T_r$ is defined such that it takes values close to zero with nonzero probability. This property allows us to set $T_0$ to the instant when the first nucleus initiates transcription, in order to determine $T_r$. The random variable $T_r$ is modeled using a finite state, continuous time, Markov chain. The states of the process are $A_1, A_2,..., A_{n-1}, A_n$. The states $A_i$, $1 \le i \le n - 1$ are OFF, i.e., not transcribing. The state $A_n$ is ON, i.e., transcribing. Each OFF state has a given lifetime, defined as the waiting time before leaving the state and going elsewhere. Like in [31], we considered that each of the states has the same lifetime denoted $\tau$. Also, the transitions are considered linear and irreversible: in order to go to $A_{i+1}$ one has to visit $A_i$ and once there, no return is possible. The time $T_r$ is the time needed to reach $A_n$ starting from one of the OFF states. The predictions of these models were compared to the empirical survival function $S_{exp}(t)$ defined as the probability that $T_r > t$, obtained using the Meier–Kaplan method from the values $T_r$ for all the analyzed nuclei. Following Occam's razor principle, we based our analysis on the simplest model that is compatible with the data, which is a model with $n = 4$ and homogeneous lifetimes (see Supplementary Fig. 8b). As in ref. [31], the number of states can be determined directly from the variance V and mean M of postmitotic reactivation time. If $a = M^2/V$, then the estimated number of states is $N = a + 1$.

For this model, the theoretical survival function is a mixture of Gamma distributions:

$$S(t) = p_1 \exp(-t/\tau) + p_2 \left(1 - \frac{1}{\Gamma(2)}\gamma(2, t/\tau)\right) + p_3 \left(1 - \frac{1}{\Gamma(3)}\gamma(3, t/\tau)\right) \qquad (10)$$

where $\gamma$, $\Gamma$ are the complete and incomplete gamma functions and $p_1$, $p_2$, $p_3$ (satisfying $p_1 + p_2 + p_3 = 1$) are the probabilities to reach ON after one, two or three jumps, respectively.

We have also tested more complex models, with uneven lifetimes, more states, and therefore, more parameters. However, the complex models did not provide a sensibly better fit with data and generated overfitting identified as large parametric uncertainty.

The parameters "a" and "b", summarizing the statistics of the postmitotic reactivation are defined as

$$a = p_1 + 2p_2 + 3p_3, \; b = \tau \qquad (11)$$

The uncertainty intervals shown in Supplementary Fig. 8c indicate the parameter sensitivity as well as the degree of constraint imposed by the cost function (which is the sum of squares of differences between the data and the fit). An uncertainty interval approaching zero implies that the parameter is very sensitive and that altering the parameter value will have a strong influence on the cost function. The frequentist approach applied here does not generally have a mechanism to determine statistical confidence intervals for a given parameter.

The goodness of fit is given by the sum of squares distance O, representing the value of the cost function for optimal parameters.

**Statistics and reproducibility**. Data represented in Fig. 1a, c (right panel); 4b and in Supplementary Fig. 1a–c; 2a, b (lower panel); 5b; 6a, b, 6g (right panel); 7c, from immunostaining and/or smFISH experiments were performed on hundreds of embryos and at least twice at two independent times.

**Reporting summary**. Further information on research design is available in the Nature Research Reporting Summary linked to this article.

## Data availability

All relevant data supporting the key findings of this study are available within the article and its Supplementary Information files or from the corresponding author upon reasonable request. All ChIP-seq data and bed files are accessible at GEO Accession viewer under the number GSE180812. The updated version of MitoTrack is available at https://github.com/ant-trullo/MitoTrack_v4_0. The distance measurement software is available through this link: https://github.com/ant-trullo/DNA_FishAnalyzer.

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

## Acknowledgements

We are grateful to G.Cavalli, B. Schuttengruber, F. Juge, F.Bantignes, E. Bertrand, and J.Chubb for sharing antibodies, probes, advise on DNA FISH and/or insightful discussions. We thank members of the Lagha lab, Cyril Esnault, Virginia Pimmett, and Etienne Schwob for their critical reading of the manuscript. We acknowledge M. Dejean and M. Goussard for technical assistance. We acknowledge the Montpellier Ressources Imagerie facility (France-BioImaging). M.B. is a recipient of an FRM fellowship. This work was supported by the ERC SyncDev starting grant to M.L. M.L., J.D., and C.F. are sponsored by CNRS. O.R. acknowledges support from the French National Research Agency (ANR-17-CE40-0036, project SYMBIONT).

## Author contributions

M.L. conceived the project. M.L., M.B., and J.D. designed the experiments. J.D., M.B., H.L.-H., M. Lam., and H.F.-G. performed experiments. A.T. developed software. C.F., M.B., and J.D. performed kinetic analysis. M.G. and M.H. provided the GAF-GFP strain. O.R. performed mathematical modeling, machine learning, and interpreted the data. G.H., M.B., A.Z.-E.-A., M.M., and J.-C.A. performed the bioinformatics analysis. M.L. and M.B. wrote the manuscript. All authors discussed, approved, and reviewed the manuscript.

## Competing interests

The authors declare no competing interests.
