## [Peer Review File · Nature Communications]

The control of transcriptional memory by stable mitotic bookmarkingReviewers' Comments:

Reviewer #1:

Remarks to the Author:

Review of "Control of transcriptional memory by stable mitotic bookmarking", Bellec et al.

This manuscript examines the behavior and effect of pioneer-like transcription factor GAF on mitotic bookmarking in *Drosophila* embryos. The authors found that GAF is retained during mitosis through mitotic immunostaining and GFP-tagged GAF imaging. Fluorescence Correlation Spectroscopy and Fluorescence Recovery After Photobleaching were used to examine GAF-GFP binding in interphasic embryos. This revealed minute-long GAF binding attributed to sequence specificity. The authors then used cell sorting by mitotic marker H3S10ph to separate mitotic embryos for the purpose of downstream ChIP. A set of mitotically retained GAF targets was identified from ChIP data. The authors examined the chromatin accessibility of GAF-bound loci with publicly available ATAC-seq data, and found that mitotic retained GAF targets were more open than GAF interphase or mitotic-only targets, suggesting a role for GAF in fostering chromatin accessibility. The authors then examined post-translational chromatin modifications in GAF retained regions. An anti-correlation between H3K27Ac and H3K27Me3 was observed in a significant number of targets. The role of GAF in mitotic looping was examined via DNA FISH, but showed no difference compared to a negative control, and so GAF was determined to have no role in mitotic looping for those loci. The authors performed quantitative imaging on a GAF mitotic target, the gene *scylla*. Post-mitotic activation of *scyl* was observed between 7.5 and 9 minutes, and showed a transcriptional memory activation bias (faster post-mitotic activation). This was found to not be related to a previously active progenitor cell bias. Lastly, GAF transcriptional memory was tested with RNAi knock-down of GAF. Post-mitotic reactivation of *scyl* showed a ~6 min delay in RNAi-GAF embryos, but no significant reduction in transcription site intensity. Lastly, a mixed gamma distribution was used to model GAF driven transcriptional memory. This model identified that the primary driver of GAF transcriptional memory at *scyl* is 'b', which is the duration of rate-limiting transitions prior to reaching the transcriptionally active state. The authors synthesize their results into a model in which loci are bookmarked by GAF, creating a distinct epigenetic path which leads to transcriptional memory bias.

Overall, the authors use a variety of analytical approaches to assess the role of GAF and transcriptional memory in *drosophila* embryos. Their conclusions are generally well supported by their data, and are within reason for the results shown. The claims of GAF mitotic bookmarking were thoroughly examined via ChIP-seq, publicly available ATAC-seq data, single-gene imaging, and RNAi experiments.

I do have some suggestions that should be addressed prior to publication:

Major suggestions:

Modeling – Should show sample model fit to experimental data somewhere in main figures or supplemental figures. Also, nice to show fit with fewer states ($n < 4$) as well as with $n = 5$, to better justify the use of $n = 4$.

Line 213-215, the reasoning behind the smiFISH experiments is unclear, as shown in Fig. 4b, Sup. Fig 6a-b. To ensure knockin at the proper location, a stronger control would be to perform a dual smiFISH experiment using probes against MS2 and *scyl* lit up with two different colors in the established cells (*Scylla_24XMS2*) and observe their colocalization in the nc13 & nc14 stages. Or, more simply, do PCR to test that the insertion site is correct.

Line 231, When it is stated that a transcriptional memory bias is observed, please quantify or justify significance. Related, error bars or confidence intervals are needed in Fig. 4d, Fig. 5b, and Sup Fig.

7e.

Fig. 1b: It would be beneficial if a snapshot of the movie or scheme like in Fig. 1c was shown to explain where the blue (His2AV), and particularly the green (GAF-GFP DNA) and the red (GAF-GFP cytoplasm) intensity signals are extracted from the cells in movie 1. It's not clear how the cytoplasmic GAF-GFP intensity signal is extracted from these images. The color code here and in movie1 could be misleading based on the "RFP & GFP" signals. One might expect to see these signals represented in red and green colors, respectively. Stick to one abbreviation format: are Histone, H2Av-RFP, and His2Av-mRFP the same? Also, it is not explained in the legends, intro, or results in text what "Histone, H2Av-RFP, and His2Av-mRFP" means. Does the shadow represent the SEM or SD?

Minor suggestions:

Make sure all acronyms are introduced at first use. E.g.s., in the text: Line 104, H3S10ph; Line 113, Snail (sna); Line 200, snail and escargot; Line 243, KD; Line 394, scyl; Line 349, 400, 507, H2Av-RFP; Line 521, RPL32.

Lines 119-120 – Awkward sentence with 4 commas, should be reworded or split for clarity.

Lines 263-264 – Typo "Prior to reach" -> "Prior to reaching"

Line 211, Nice to cite the original paper in which the MS2/MCP system was first employed (Bertrand et al. 1998).

Paragraph 232-236, I don't think it is mentioned in the methods how the quantification of the transcription site spots is carried out?

Line 218, (movie 3) is referred to; however, the legend mentions mat-alpha-Gal4 and nos-Gal4, when these terms have not been introduced.

Lines 264-267, was the model applied to the intensities or the % of activation in the transcriptional dynamics' movies? Please clarify.

Line 791, Indicate the laser power in mW used.

Line 831, Typo in the 1st word.

Lines 883-884, refer to the paper in which FLAP probes were originally proposed (Tsanov et al. 2016). Which type of FLAP probes were used in this study, X,Y, or Z?

Line 891, How many z-planes were taken? Did this number of Z-stacks cover the cells from top to bottom?

Line 964, Why were these thresholds (100 & 22) selected?

Lines 982-983, Ramirez et al citation is not shown as a superscript, like other citations.

Lines 987-1014, Live imaging section, Include the laser power and exposure times used for each movie.

Line 994, Do you mean tracking was performed on maximum intensity projection images? It is a little confusing as written.

Lines 1019, is the custom-made algorithm available somewhere?

Lines 1024-1027, explain how the intensity was quantified? Which method was used? Was the background subtracted?

Line 1043, add min to 1h10...

Line 1047, should be "antibodies", and change hour to h to stick to the same format.

Line 1074, How many z-stacks were taken in total?

Fig. 1a: How many experimental replicates showed the same effect presented in Fig. 1a.? (It doesn't say in the figure legend)

Fig. 1f: Should mention a reaction-diffusion model was used to fit for K_{off} and D_{eff} in the main text here.

Sup. Fig. 1 a-b: How many biological replicates showed the same effect presented in these figures?

Fig 2a – Description is inadequate, should be an in-depth description.

Fig. 2c: It is not clearly explained why *ca-beta* & *muc11a* were chosen? I understand they are representative for interphase-only and mitosis-only, but why were those genes selected in particular?

Fig. 3 a: y-axis units?.

Fig. 3 a, c-d: The n numbers are not explained here.

Fig 3b – Statistical significance test for difference between GAF and control peaks? Hard to trust the difference without one, and no error is shown. Can this difference be quantified, and the quantification related to the actual difference in chromatin accessibility?

Fig. 4 a: Describe a little bit more the schematic to explain better the strategy used for CRISPR editing.

Fig. 4b: Is the purpose of the smiFISH experiment using MS2 probes to confirm that the *scyl* endogenous gene was successfully edited and MS2 was inserted? If so, describe it better in the text (paragraph 207-218), so it is more explicit what exactly is being compared.

Sup. Fig. 2a: How many biological replicates showed the same effect presented in these figures? Is the brightness / contrast scaling the same for all shown images? H3S10ph is still present in interphase cells, although dimly. The contrast between interphase / mitotic images would suggest two discrete populations, however, the cell sorting shows an even and linear distribution. The cell-sorting windows are conservative enough that this should not affect downstream results, however. Note that Y-axis also needs a label.

Sup. Fig. 3b,c: Keep y-axis same to enable comparisons

Sup. Fig. 3e: What are the criteria to select *tkv*, *disco-r*, and *mef2* as representative of each cluster?

Sup. Fig. 5: What targets were used/selected as a control region for the distance experiments when comparing the *scyl* and *chrB* genes, and *esg* and *sna* genes? and why?

Sup. Fig. 7a: Whole blots should be displayed for Western Blot experiments. Quantification of the protein amounts obtained from the western blot experiments should be shown.

Sup. Fig. 7c: There are not error bars before 5 minutes?

Sup. Fig. 7d: Why are there no visible error bars for some of the bars? Is this because there are so small? I suggest showing all fitted values for this measurement.

Movie legends: Overall, the description of all the movies should be expanded, and not copied & paste (e.g. movie 3 & 4 legends). It should include a clear description of everything in the movie.

Movie 4 seems to be overprocessed for the MS2 channel, and spurious spots seem to be displaying.

Reviewer #2:

Remarks to the Author:

In the manuscript "The control of transcriptional memory by stable mitotic bookmarking", the authors reveal that pioneer-like transcription factor GAF acts as stable mitotic bookmarker during zygotic genome activation. The authors further discovered that a large fraction of GAF mitotic targets remains associated with GAF during interphase. In addition, the authors annotated those GAF target sites using H4K8ac ChIP-seq data and other publicly available epigenetic data to connect GAF binding to the chromatin landscape.

Overall, the authors provided interesting results to dissect a new regulatory role of GAF in early drosophila embryo development. The generated GAF and H4K8ac ChIP-seq data are valuable resources to the community (they already uploaded the data to the GEO database). Bioinformatics data analysis aligns well with the current standard and supports their claims. However, I do have a few comments related to bioinformatics for the authors to address.

1) Please include the called peak regions (traces for bed files) in Figure 2c, c', c''.

2) Figure 2f, f', f'' Can the authors include some statistical analysis to show the significance of difference between interphase only and mitotically retained? Can the authors try HOMER to annotate the peak regions (tss, tts, exon, intron, intergenic, etc)? Can the authors calculate the genomic distance from the peak center to the nearest gene tss and generate distribution plots for different categories of GAF targets? In addition, I wonder whether known and de novo motif analysis using HOMER can provide additional insight on enrichment of other TF motifs for different categories of GAF targets?

3) Suppl. Figure3d showed 180 (96%) GAF level-dependent accessible chromatin regions overlap with GAF ChIP-seq target regions identified as mitotically retained. Are the 180 peak regions overlap with the top hits of mitotic GAF targets with the strongest ChIP signals? More importantly, how many of the 180 peak regions overlap with TSS, enhancers, TAD boundaries, and other regions. Additional statistical analysis could help.

4) Figure3b, Suppl. Figure3b, 3c: can the authors add Kolmogorov-Smirnov test or some other statistical test to compare the curves? I assume that in Suppl. Figure3c Mitotically retained – TAD boundaries, GAF_control and GAF_degradFP curves are not significantly different, is that correct?

Reviewer #3:

Remarks to the Author:

The authors address the role of GAF in mitotic bookmarking in the Drosophila embryo. They provide evidence for stable GAF binding and use mitotic ChIP to identify targets that retain GAF binding during

mitosis. Subsets of these GAF targets have different chromatin marks. The authors use MS2 live imaging of the GAF mitotically retained target gene *scylla* in single nuclei to provide evidence that memory is disrupted upon GAF depletion. Overall, the data are of high quality but one issue is that population level data on binding and chromatin modifications are correlated with single cell transcriptional responses, which makes interpreting the data more difficult.

Major comments

It is not clear to me exactly what the authors model is. I think it is that GAF can be bound pre/during mitosis and facilitate chromatin changes allowing rapid activation that is attributed to transcriptional memory. But what do the authors think is happening in the nuclei that become active from inactive mothers? Is it that they assume GAF is bound in the interphase to allow activation (with a lag) or is it that the gene of interest (eg *scylla*) is activated in the absence of GAF binding? I think the data presented are more supportive of the former. The results section ends with a line about stochastic GAF binding but this is never elaborated on. It would help the reader to discuss more fully what the data show and what the shortcomings are in the discussion.

From the text description, it seems that the conclusion from the data shown in Sup Fig 1b is that the large GAF puncta are due to heterochromatin binding apically and the more basal uniform GAF signal represents binding to euchromatin. Is this right? In Fig S1 at *nc14* there appears to be poor colocalization between the peri-centromeric DNA regions tested and GAF. Could it not be that the GAF puncta are associated with transcriptional activation as in a hub model? The GAF puncta seem interesting but are barely discussed.

The conclusion that there is stable binding of GAF to chromatin, including during mitosis, is central to the paper, yet the supporting evidence is weak. What is the evidence that the short and long recovery times from the FRAP experiment (Fig 1d) correspond to diffusion and stable binding, respectively? Studying a GAF DNA binding mutant would strengthen this conclusion. Were the FRAP and FCS measurements taken apically or basally? It would be interesting to measure the stability of GAF DNA interaction in heterochromatin vs euchromatin, if that really is what is represented apically and basally, as there may be differential stability. I am also confused as to how the data in Fig 1e relates to Fig S1e, f.

In Fig 3, clusters 2 and 3 seem to be transcriptionally inactive – can the authors speculate what these subpopulations of repressed genes might represent? Also, the embryo timings for the data shown in Fig 3/Fig S3 are not clear.

The MS2 data (Fig 4/Fig S6) show that only a proportion of nuclei are active in *nc13* and *nc14*. Is it the case that typically only one daughter is active following division? I think the data in Fig S6e shows that at most 50% of the daughters from *nc13* active nuclei are also active in *nc14* (but I may have misunderstood it, I found the legend confusing and it is not clear if the data relate to dorsal or ventral nuclei). Is there any difference in the ability of both daughters to activate transcription depending on whether the mother was active or inactive? Linked to this, it would be interesting if the authors could explain how they think bookmarking works molecularly – is it that both replicated chromosomes have GAF bound and modified chromatin or only one?

GAF depletion slows the activation in daughter nuclei from both active and inactive nuclei. Does the statistically significant delay in activation observed in daughters from inactive mother nuclei in the presence of GAF RNAi (relative to inactive mothers with white RNAi) not mean that GAF has a memory independent role in activation? The key question is whether the similar activation time observed in daughters from active GAF RNAi and inactive white RNAi mothers is due to a loss of memory in the active mothers or simply a coincidence based on overall slower activation profiles. I did not find the argument that the effect of GAF RNAi on daughters from active mothers is due to a loss of memory completely convincing.

What do the authors think is the basis for the stochastic activation from inactive mothers? Could it be that GAF is only bound in a subset of the cells post-mitosis and for those without bound GAF the delay reflects a requirement for GAF recruitment? If this is correct, activation in daughters from inactive (but not active) mothers might depend on a high local GAF concentration, for example proximity to the GAF puncta. This should be testable by imaging GAF-GFP and MS2 (with MCP-RFP) simultaneously in nuclei.

Can the authors speculate as to what they think the different states represent in their model?

Minor comments

How do the authors know that the GAF-GFP being measured in the nucleus is on DNA in the interphases, as indicated in Fig 1b?

In Fig 2f – what does the 'other' category for the mitosis only class reflect? Is it mostly gene bodies or intergenic regions?

Fig S3a shows that a higher number of GAGAG motifs is associated with a wider region of more weakly accessible chromatin. What might the reason for this be?

For the cartoon in Fig 4d – should the green nucleus at nc13 not be depicted as transcriptionally active?

Point by point answer to reviewer's comments

The control of transcriptional memory by stable mitotic bookmarking

Maëlle Bellec¹, Jérémy Dufourt¹, George Hunt², Hélène Lenden-Hasse¹, Antonio Trullo¹, Amal Zine El Aabidine¹, Marie Lamarque¹, Marissa M Gaskill³, Heloïse Faure-Gautron¹, Mattias Mannervik², Melissa M Harrison³, Jean-Christophe Andrau¹, Cyril Favard⁴, Ovidiu Radulescu⁵, and Mounia Lagha¹.

Color code for the Point-by-point answer:

Reviewer comments are in italics.

Our answers are in blue. Quotes from the revised text are in green.

We are very grateful to the reviewers for their various comments that were extremely helpful in improving our manuscript. We did our best to incorporate the majority of these excellent suggestions. Below we provide a detailed, point-by-point account of the changes in the revised manuscript.

The most notable revisions include new bioinformatic analyses, clarifications of the modeling aspect, new FCS data and new experiments questioning the link between GAF hubs and transcriptional activation. The revised manuscript now contains 8 Supplementary Figures, 6 Movies (Movies 3 and 4 are new) and 4 Tables (Table 4 is new), as well as 2 Tables and 8 Figures for the referees included in this rebuttal.

Reviewer #1 (Remarks to the Author):

This manuscript examines the behavior and effect of pioneer-like transcription factor GAF on mitotic bookmarking in Drosophila embryos. The authors found that GAF is retained during mitosis through mitotic immunostaining and GFP-tagged GAF imaging. Fluorescence Correlation Spectroscopy and Fluorescence Recovery After Photobleaching were used to examine GAF-GFP binding in interphasic embryos. This revealed minute-long GAF binding attributed to sequence specificity. The authors then used cell sorting by mitotic marker H3S10ph to separate mitotic embryos for the purpose of downstream ChIP. A set of mitotically retained GAF targets was identified from ChIP data. The authors examined the chromatin accessibility of GAF-bound loci with publicly available ATAC-seq data, and found that mitotic retained GAF targets were more open than GAF interphase or mitotic-only targets, suggesting a role for GAF in fostering chromatin accessibility. The authors then examined post-translational chromatin modifications in GAF retained regions. An anti-correlation between H3K27Ac and H3K27Me3 was observed in a significant number of targets. The role of GAF in mitotic looping was examined via DNA FISH, but showed no difference compared to a negative control, and so GAF was determined to have no role in mitotic looping for those loci. The authors performed quantitative imaging on a GAF mitotic target, the gene scylla. Post-mitotic activation of scyl was observed between 7.5 and 9 minutes, and showed a transcriptional memory activation bias (faster post-mitotic activation). This was found to not be related to a previously active progenitor cell bias. Lastly, GAF transcriptional memory was tested with RNAi knock-down of GAF. Post-mitotic reactivation of scyl showed a ~6 min delay in RNAi-GAF embryos, but no significant reduction in transcription site intensity. Lastly, a mixed gamma distribution was used to model GAF driven transcriptional memory. This model identified that the primary driver of GAF transcriptional

memory at scyl is 'b', which is the duration of rate-limiting transitions prior to reaching the transcriptionally active state. The authors synthesize their results into a model in which loci are bookmarked by GAF, creating a distinct epigenetic path which leads to transcriptional memory bias.

Overall, the authors use a variety of analytical approaches to assess the role of GAF and transcriptional memory in drosophila embryos. Their conclusions are generally well supported by their data, and are within reason for the results shown. The claims of GAF mitotic bookmarking were thoroughly examined via ChIP-seq, publicly available ATAC-seq data, single-gene imaging, and RNAi experiments.

I do have some suggestions that should be addressed prior to publication:

We thank Referee1 for his/her in-depth reading of our manuscript. We did our best to address all of his/her comments in the detailed point-by-point answer below.

Major suggestions:

1.1) Modeling– *Should show sample model fit to experimental data somewhere in main figures or supplemental figures. Also, nice to show fit with fewer states ($n < 4$) as well as with $n = 5$, to better justify the use of $n = 4$.*

We have also followed the referee's advice and fitted with fewer ($N < 4$) and with $N = 5$ states. The results are presented in Supplementary Figure 8b. As expected, the accuracy of the fit is improved when N is increased. However, the improvement from $N = 4$ to $N = 5$ is negligible. By parsimony, we can stop at $N = 4$. A sentence has been added to the method section to illustrate this point (lines 1249-1250).

1.2) Line 213-215, *the reasoning behind the smiFISH experiments is unclear, as shown in Fig. 4b, Sup. Fig 6a-b. To ensure knockin at the proper location, a stronger control would be to perform a dual smiFISH experiment using probes against MS2 and scylla lit up with two different colors in the established cells (Scylla_24XMS2) and observe their colocalization in the nc13 & nc14 stages. Or, more simply, do PCR to test that the insertion site is correct.*

To answer to this point, we performed RNA FISH labeling with *scylla* and MS2 probes in *scylla_24XMS2/+* embryos. We added a new panel in Supplementary Figure 6a and commented in the text line 243. We observed a similar pattern between *scylla* and MS2 probes suggesting that the expression domain of *scylla* is not affected by the insertion of MS2 loops. We also see that the transcription site of the *scylla_24XMS2* allele (labeled with MS2 probes) perfectly co-localizes with one of the two transcription foci of the wild type *scylla* allele (without MS2 sequences).

We think that the new smiFISH data and the fact that homozygous *scylla_24XMS2/scylla_24XMS2* strain is viable and healthy confirm that the insertion of the MS2 loops do not interfere with the expression of the *scylla* gene. A sentence regarding this point has been added line 243.

1.3) Line 231, When it is stated that a transcriptional memory bias is observed, please quantify or justify significance. Related, error bars or confidence intervals are needed in Fig. 4d, Fig. 5b, and Sup Fig. 7e.

In the figure legend of Fig.4, Fig.5 and Sup Fig7, we clarified how transcriptional memory was quantified. In brief, nuclei are pooled from several movies (4 different movies), and their timing of first activation post-mitosis is plotted as a cumulative activation curve. We show the timing required to 'fill' a given pattern, here the central part of the mesoderm. With this type of representation, it is difficult to show error bars or statistical tests as each movie do not contain enough data to be considered as an independent experiment. However, in Figure 5c, for which we represented the time of activation of each nucleus in a bar plot, we used a statistical test (t-test) to show the significant difference.

1.4) Fig. 1b: It would be beneficial if a snapshot of the movie or scheme like in Fig. 1c was shown to explain where the blue (His2AV), and particularly the green (GAF-GFP DNA) and the red (GAF-GFP cytoplasm) intensity signals are extracted from the cells in movie 1. It's not clear how the cytoplasmic GAF-GFP intensity signal is extracted from these images. The color code here and in movie1 could be misleading based on the "RFP & GFP" signals. One might expect to see these signals represented in red and green colors, respectively. Stick to one abbreviation format: are Histone, H2Av-RFP, and His2Av-mRFP the same? Also, it is not explained in the legends, intro, or results in text what "Histone, H2Av-RFP, and His2Av-mRFP" means. Does the shadow represent the SEM or SD?

We agree with the referee that annotations can be misleading. We therefore changed the color of the graph in Figure 1b. We also changed the color of the Movie 1 according to referee's suggestion and the color of the Supplementary Figure 6b for consistency. In Figure 1b, shadows are SEM and this is now specified in the figure legend. We also replaced 'GAF-GFP on DNA' by 'GAF-GFP in nucleus' in the legend of panel 1b. For simplicity, we kept 'Histone' for the red curve and specify it in the legend line 386.

Minor suggestions:

Make sure all acronyms are introduced at first use. E.g., in the text: Line 104, H3S10ph; Line 113, Snail (sna); Line 200, snail and escargot; Line 243, KD; Line 394, scyl; Line 349, 400, 507, H2Av-RFP; Line 521, RPL32.

This is now corrected.

Lines 119-120 – Awkward sentence with 4 commas, should be reworded or split for clarity.

We replaced the sentence "Thus, we established a pipeline, able to profile mitotic nuclei at a genomic scale, for the first time in a multicellular organism, in the absence of drug synchronization" by 2 sentences (line 144).

"Thus, we established a pipeline able to profile mitotic nuclei at a genomic scale in the absence of drug synchronization. To our knowledge, this protocol and data represent the first natural mitotic chip performed in a multicellular organism."

Lines 263-264 – Typo "Prior to reach" -> "Prior to reaching"

Corrected, thanks for pointing out this typo.

Line 211, Nice to cite the original paper in which the MS2/MCP system was first employed (Bertrand et al. 1998).

We thank the reviewer for pointing this. We have now added the reference to Bertrand et al. 1998 for the MS2/MCP technique, line 50 and line 241.

Paragraph 232-236, I don't think it is mentioned in the methods how the quantification of the transcription site spots is carried out?

The detection and the timing of transcription sites was quantified as in Trullo et al. 2019. We now added a paragraph in the method section describing more in-depth the quantification of the timing and the intensities of each transcription site (lines 1143-1152).

Line 218, (movie 3) is referred to; however, the legend mentions mat-alpha-Gal4 and nos-Gal4, when these terms have not been introduced.

We apologize for not being specific enough. We have added more description in the movie legend (now Movie 5 and Movie 6), line 862 and 870.

Lines 264-267, was the model applied to the intensities or the % of activation in the transcriptional dynamics' movies? Please clarify.

The modeling was applied to the waiting time between mitosis and first activation. Modeling did not consider the intensities of transcription spots. We have clarified this in the main text line 295.

Moreover, to avoid confusion, we removed the panel depicting transcription site intensities in *RNAi white* and *RNAi GAF* embryos (old Figure Sup 7d). Indeed, decoding the impact of GAF on transcriptional bursting is outside the scope of this work, primarily based on timing of activation (and not on levels). Accordingly, we deleted the 2 sentences referring to this panel in the main text line 284.

Line 791, Indicate the laser power in mW used.

The laser power of the 488 laser for FRAP acquisition images is 5uW. Measurements are taken with a 10X objective. Moreover, our FRAP data were normalized for photobleaching. This is now included in the method section line 918.

Line 831, Typo in the 1st word.

Corrected. Thanks.

Lines 883-884, refer to the paper in which FLAP probes were originally proposed (Tsanov et al. 2016). Which type of FLAP probes were used in this study, X, Y, or Z?

We thank the referee for pointing this and now specify the use of the FLAP-Y probes line 1004.

Line 891, How many z-planes were taken? Did this number of Z-stacks cover the cells from top to bottom?

Our imaging settings use a number of Z-planes that cover the entire depth of the cells, in order to image cells from top to bottom. For each image of all figure, the number of Z-stacks is indicated in the methods (e.g., line 991).

Line 964, Why were these thresholds (100 & 22) selected?

To call the enriched peaks from the final wiggle files, we used Thresholding function of the Integrated Genome Browser (IGB) to define the signal value over which we consider a genomic

region to be enriched compared to background noise (Threshold). The values 100 and 22 were chosen regarding what we considered as a good signal-to-noise ratio.

Lines 982-983, Ramirez et al citation is not shown as a superscript, like other citations.

This is now corrected. Thanks.

Lines 987-1014, Live imaging section, Include the laser power and exposure times used for each movie.

These settings are now included in the live imaging section (lines 1123, 1131 and 1141). Exposure time cannot be included as we are using a confocal microscope with a PMT and not a camera.

Line 994, Do you mean tracking was performed on maximum intensity projection images? It is a little confusing as written.

We clarified the sentence line 1118, as such “Maximum intensity projected images were used for automatic tracking using a home-made software as in ¹.”

Lines 1019, is the custom-made algorithm available somewhere?

The algorithm is available at: https://github.com/ant-trullo/MitoTrack_v4_0.

The link is now provided in the method section line 1157.

Lines 1024-1027, explain how the intensity was quantified? Which method was used? Was the background subtracted?

We apologize if our original explanations were unclear. We have now added a paragraph in the method section to precisely describe how intensities were quantified (line 1152).

Line 1043, add min to 1h10...

Corrected.

Line 1047, should be “antibodies”, and change hour to h to stick to the same format.

Corrected.

Line 1074, How many z-stacks were taken in total?

For DNA FISH experiments, imaging must cover the entirety of nuclei depth (to avoid missing a DNA spot). As nuclei elongate during time, the number of z-planes required to cover them is variable and scales with the embryonic stage. Therefore, the number of z-planes was not fixed, although the spacing of the z-planes remained constant between images. A typical image is 40-60 z-planes.

Fig. 1a: How many experimental replicates showed the same effect presented in Fig. 1a.? (It doesn't say in the figure legend)

A typical immuno-fluorescence experiment is performed on hundreds of embryos and at least in two independent experiments. Usually almost the entire slide is visually checked to see the effect and in that case 3 different embryos at each stage were imaged.

Fig. 1f: Should mention a reaction-diffusion model was used to fit for Koff and Deff in the main text here.

We added this information in the main text (line 107) as well as a better explanation of FCS and FRAP experimental interpretations (see point 3.5).

Sup. Fig. 1 a-b: How many biological replicates showed the same effect presented in these figures?

A typical immuno-fluorescence experiment is done on hundreds of embryos and at least in two independent experiments. Usually almost the entire slide is visually checked to see the reproducibility of the effect. For the results shown in Sup Fig1. a-b, 5 different embryos were imaged.

Fig 2a – Description is inadequate, should be an in-depth description.

We added a more detailed description to the Figure legend of Fig 2a line 401.

Fig. 2c: It is not clearly explained why ca-beta & muc11a were chosen? I understand they are representative for interphase-only and mitosis-only, but why were those genes selected in particular?

We arbitrary selected representative GAF targets for each category.

Fig. 3 a: y-axis units?

The y-axis has been added. Thanks for pointing out this omission.

Fig. 3 a, c-d: The n numbers are not explained here.

n represents the number of identified GAF peaks. This is now mentioned in the figure legend of Figure 3 line 422.

Fig 3b – Statistical significance test for difference between GAF and control peaks? Hard to trust the difference without one, and no error is shown. Can this difference be quantified, and the quantification related to the actual difference in chromatin accessibility?

We agree with the referee and performed a Wilcoxon test to compare the curves of ATAC-seq accessibility in *GAF_control* and *GAF_degradFP* (also suggested by referee 2, see point 2.4). We performed this test on +/-500bp around the center of GAF peaks, and the p-value was <0.01 for mitotically retained GAF peaks but not for interphase only or Zelda only peaks. It is now specified in the Figure 3b and legend.

Fig. 4 a: Describe a little bit more the schematic to explain better the strategy used for CRISPR editing.

This is now specified in the Figure 4a legend.

Fig. 4b: Is the purpose of the smiFISH experiment using MS2 probes to confirm that the scyl1 endogenous gene was successfully edited and MS2 was inserted? If so, describe it better in the text (paragraph 207-218), so it is more explicit what exactly is being compared.

Please refer to the answer to Referees comment 1.2. We performed single molecule inexpensive RNA FISH experiments (smiFISH) with MS2 and *scylla* probes in *scyl1_24XMS2/+* embryos. As shown in the representative image Supplementary Figure 6a, MS2 and *scyl1* mRNA patterns largely overlap.

Sup. Fig. 2a: How many biological replicates showed the same effect presented in these figures? Is the brightness / contrast scaling the same for all shown images? H3S10ph is still present in interphase cells, although dimly. The contrast between interphase / mitotic images would suggest two discrete populations, however, the cell sorting shows an even and linear distribution. The cell-sorting windows are conservative enough that this should not affect downstream results, however. Note that Y-axis also needs a label.

We did such immunostaining on hundreds of embryos and observed this effect on tens of them. The contrast scaling is the same for all images.

Here the referee questions the shape of the sorting profile. Indeed, while the images suggest discrete populations, the sorting profile exhibits a linear distribution. This can be due to: 1) the difference of intensity between the various phases of mitosis; and/or 2) the non-complete synchronicity of mitosis in a subset of embryos, as shown in representative images in Sup2b.

Embryo sorter is not a highly sensitive fluorescence sorter and is hard to set up, this is why each sorting were visually inspected to control the goodness of the sorting.

Sup. Fig. 3b,c: Keep y-axis same to enable comparisons

The goal of this figure is not to compare profiles between different genomic locations (eg. TSS or Enhancers) but rather to contrast ATACseq profiles within a category between two genetic backgrounds (GAF control versus GAF_degradFP for Sup Fig3c or mitosis versus interphase for Sup Fig3b). We refrained from keeping the same y axis as these data originate from different studies (Blythe and Wieschaus 2016 for panel b and Gaskill et al. 2021 for panel c).

Sup. Fig. 3e: What are the criteria to select tkv, disco-r, and mef2 as representative of each cluster?

These genes were selected arbitrarily, to illustrate representative targets (with clear enrichment of GAF ChIP-seq signal) for each cluster group.

Sup. Fig. 5: What targets were used/selected as a control region for the distance experiments when comparing the scyl and chrB genes, and esg and sna genes? and why?

Control regions were chosen to be at equivalent distance from *chrB* or *sna* promoters, as we know that genomic distance impacts contact frequencies⁴. Moreover, in⁵⁵, the authors used the same control at equi-genomic distance and found a closer proximity of *scyl-chrB* compared to *chrB-control*.

Sup. Fig. 7a: Whole blots should be displayed for Western Blot experiments. Quantification of the protein amounts obtained from the western blot experiments should be shown.

We now provide whole blots (Supplementary Figure 7b), and added a related quantification.

Sup. Fig. 7c: There are not error bars before 5 minutes?

There are no errors bars before 5min because the intensity is from too few nuclei at these early time points.

Sup. Fig. 7d: Why are there no visible error bars for some of the bars? Is this because there are so small? I suggest showing all fitted values for this measurement.

We thank the reviewer for pointing this. Error bars do not represent error in the fitting but correspond to variation among optimal and best suboptimal fits. We now added this description in the figure caption of the Figure 7d line 809.

Furthermore, the referee point raised the need to include a table with the different value of the parameters of the mathematical modeling. The table of parameters is now provided in Sup Table 4.

Finally, we added a section in the method (lines 1263-1267) to describe more precisely how the uncertainty of intervals are assessed.

Movie legends: Overall, the description of all the movies should be expanded, and not copied & paste (e.g. movie 3 & 4 legends). It should include a clear description of everything in the movie.

We now provide a better description of the movie in the movie legends.

Movie 4 seems to be overprocessed for the MS2 channel, and spurious spots seem to be displaying.

We thank the referee for pointing out this issue; we have now replaced Movie4 (now Movie 6) with a better display.

Reviewer #2 (Expertise in the analysis of sequencing data)

In the manuscript “The control of transcriptional memory by stable mitotic bookmarking”, the authors reveal that pioneer-like transcription factor GAF acts as stable mitotic bookmarker during zygotic genome activation. The authors further discovered that a large fraction of GAF mitotic targets remains associated with GAF during interphase. In addition, the authors annotated those GAF target sites using H4K8ac ChIP-seq data and other publicly available epigenetic data to connect GAF binding to the chromatin landscape.

Overall, the authors provided interesting results to dissect a new regulatory role of GAF in early drosophila embryo development. The generated GAF and H4K8ac ChIP-seq data are valuable resources to the community (they already uploaded the data to the GEO database). Bioinformatics data analysis aligns well with the current standard and supports their claims. However, I do have a few comments related to bioinformatics for the authors to address.

We thank Referee 2 for his/her comments on the genomics aspect of the work. Thanks to his/her suggestions, we performed new bio-informatic analyses that nicely complement our original analyses.

2.1) Please include the called peak regions (traces for bed files) in Figure 2c, c', c''.

We have now included the traces for the bed files in the Figure 2c, c' and c''.

2.2) Figure 2f,f',f'' Can the authors include some statistical analysis to show the significance of difference between interphase only and mitotically retained?

In order to test the significance of the number of cis-regulatory regions in Figure 2f, f', f'', we applied a Fisher test between interphase only and mitotically retained peaks. The obtained p-value was <0.0001. This is now specified in Figure 2, panel f, f'.

Can the authors try HOMER to annotate the peak regions (tss, tts, exon, intron, intergenic, etc)?

Under the advice of the Referee, we used the HOMER algorithm to annotate GAF peak regions. This analysis is shown in Supplementary Figure 2e and is consistent with results from our analysis based on published annotations (shown in Figure 2f, f' and f''). In the HOMER analysis, we see a clear enrichment of promoter and intronic regions, probably including enhancers. This new results are mentioned in the manuscript line 161.

Can the authors calculate the genomic distance from the peak center to the nearest gene tss and generate distribution plots for different categories of GAF targets?

We calculated then distance between the center of the peak and the nearest TSS for each three categories of the peaks, mitotically retained, interphase only and mitotic only (Figure for Referee 1). We found that on average mitotically retained peaks are closer to TSS.

FIGURE 1: Violin plots of the distance of GAF peak center to the nearest TSS, in the mitotically retained, interphase only and mitotic only categories. Two tailed Welch's t-test **** $p < 0.0001$.

In addition, I wonder whether known and de novo motif analysis using HOMER can provide additional insight on enrichment of other TF motifs for different categories of GAF targets?

We ran motif search using HOMER tools and found enrichment for some known motifs as shown in the Table 1. As with the MEME motif search, we didn't find any motif enrichment for the mitotic only GAF peaks. Interestingly, it shows an enrichment for Caudal and Bicoid motif, in both categories but more prominently in the mitotically retained GAF peaks. This is most likely due to the enrichment of cis-regulatory regions in the mitotically retained category.

Mitotically retained peaks

Motif Name	Consensus	Log P-value	% of Target Sequences with Motif
Trl(Zf)/S2-GAGAFactor-ChIP-Seq(GSE40646)/Homer	RGAGAGAG	-6.219e+02	95.64%
Initiator/Drosophila-Promoters/Homer	NTCAGTYG	-1.530e-01	55.93%
Unknown6/Drosophila-Promoters/Homer	AATTTTAAAA	-2.495e+00	52.25%
caudal(Homeobox)/Drosophila-Embryos-ChIP-Chip(modEncode)/Homer	GGYCATAAAW	-1.371e-03	46.99%
bcd(Homeobox)/Embryo-Bcd-ChIP-Seq(GSE86966)/Homer	VNNGGATTADNN	-2.870e-04	33.60%
Unknown5/Drosophila-Promoters/Homer	GCTGATAASV	-4.258e+00	30.68%
Unknown2/Drosophila-Promoters/Homer	CATCMCTA	-7.454e+00	22.67%
E-box/Drosophila-Promoters/Homer	AACAGCTGTTHN	-3.956e+01	16.86%
Zelda(Zf)/Embryo-zld-ChIP-Seq(GSE65441)/Homer	KBCTACCTGW	-4.297e+00	10.97%
Dorsal(RHD)/Embryo-dl-ChIP-Seq(GSE65441)/Homer	GGGAAAAMCCCG	-2.332e+00	9.45%
Unknown4/Drosophila-Promoters/Homer	AAAAATACCRMA	-1.517e+00	9.45%
Unknown1(NR/Ini-like)/Drosophila-Promoters/Homer	MYGGTCACACTG	-5.188e+00	8.47%
M1BP(Zf)/S2R+-M1BP-ChIP-Seq(GSE49842)/Homer	CAGTGTGACCGT	-5.877e+00	8.39%
DREF/Drosophila-Promoters/Homer	AVYTATCGATAD	-2.888e+00	6.19%
Unknown3/Drosophila-Promoters/Homer	ACVAKCTGGCAGCGC	-2.385e+00	6.10%
TATA-box/Drosophila-Promoters/Homer	CTATAAAAGCSV	-1.458e+00	4.75%
dHNF4(NR)/Fly-HNF4-ChIP-Seq(GSE73675)/Homer	GGTCCAAAGTCCAMT	-1.439e+00	1.69%

Interphase only peaks:

Motif Name	Consensus	Log P-value	% of Target Sequences with Motif
Trl(Zf)/S2-GAGAFactor-ChIP-Seq(GSE40646)/Homer	RGAGAGAG	-1.392e+03	92.58%
E-box/Drosophila-Promoters/Homer	AACAGCTGTTHN	-3.397e+01	6.82%
Unknown5/Drosophila-Promoters/Homer	GCTGATAASV	-7.769e+00	13.41%
Unknown3/Drosophila-Promoters/Homer	ACVAKCTGGCAGCGC	-7.707e+00	2.25%
Dorsal(RHD)/Embryo-dl-ChIP-Seq(GSE65441)/Homer	GGGAAAAMCCCG	-4.809e+00	3.11%
Zelda(Zf)/Embryo-zld-ChIP-Seq(GSE65441)/Homer	KBCTACCTGW	-3.009e+00	4.12%
DREF/Drosophila-Promoters/Homer	AVYTATCGATAD	-2.077e+00	1.79%
dHNF4(NR)/Fly-HNF4-ChIP-Seq(GSE73675)/Homer	GGTCCAAAGTCCAMT	-1.040e+00	0.68%
Unknown6/Drosophila-Promoters/Homer	AATTTTAAAA	-8.838e-01	19.92%
Unknown1(NR/Ini-like)/Drosophila-Promoters/Homer	MYGGTCACACTG	-7.728e-01	2.30%
TATA-box/Drosophila-Promoters/Homer	CTATAAAAGCSV	-6.959e-01	1.62%
Unknown4/Drosophila-Promoters/Homer	AAAAATACCRMA	-6.302e-01	2.20%
M1BP(Zf)/S2R+-M1BP-ChIP-Seq(GSE49842)/Homer	CAGTGTGACCGT	-2.740e-01	2.15%
Unknown2/Drosophila-Promoters/Homer	CATCMCTA	-1.073e-01	8.28%
caudal(Homeobox)/Drosophila-Embryos-ChIP-Chip(modEncode)/Homer	GGYCATAAAW	-6.715e-02	19.85%
Initiator/Drosophila-Promoters/Homer	NTCAGTYG	-6.640e-02	27.30%
bcd(Homeobox)/Embryo-Bcd-ChIP-Seq(GSE86966)/Homer	VNNGGATTADNN	-1.200e-05	12.02%

Table 1: Motif enrichment of known motifs using HOMER tool, in the GAF mitotically retained and interphase only peaks.

2.3) Suppl. Figure3d showed 180 (96%) GAF level-dependent accessible chromatin regions overlap with GAF ChIP-seq target regions identified as mitotically retained. Are the 180 peak regions overlap with the top hits of mitotic GAF targets with the strongest ChIP signals?

More importantly, how many of the 180 peak regions overlap with TSS, enhancers, TAD boundaries, and other regions. Additional statistical analysis could help.

Following the referee's suggestion, we compared GAF peak intensities between those that lose accessibility upon GAF depletion (in GAF_degradFP embryos) and the pool of all GAF mitotically retained peaks (Figure for Referee 2). We found that the peaks that lose accessibility have a slightly higher signal in the GAF ChIPseq. This is in agreement with the fact that those regions are mostly cis-regulatory regions and more likely to be enriched in GAGAG motifs (see Figure for Referee 3 below).

FIGURE 2: Violin plots of the intensities of GAF peaks, of the peaks that lose accessibility upon GAF depletion (in GAF_degradFP embryos) and all mitotically retained peaks. Two tailed Welch's t-test **** $p < 0.0001$.

We looked at TSSs, enhancers and insulators in regions that lose accessibility upon GAF depletions, and found that most of these regions corresponds to TSSs and enhancers (Figure for referee 3).

FIGURE 3: Pie chart of the proportions (in percentage) in regions that lose accessibility upon GAF depletion (GAF_degradFP embryos).

2.4) *Figure3b, Suppl. Figure3b, 3c: can the authors add Kolmogorov-Smirnov test or some other statistical test to compare the curves? I assume that in Suppl. Figure3c Mitotically retained – TAD boundaries, GAF_control and GAF_degradFP curves are not significantly different, is that correct?*

As suggested by the reviewer, we did a statistical test to compare the curves of ATACseq accessibility in GAF_control and GAF_degradFP of Figure 3b and Supplementary Figure 3c and added the p-values on the corresponding figure. We chose a Wilcoxon test as we wanted to compare the median values of the curves, at +/-500bp from the center of the peaks. We thank the referee for this suggestion as it clearly shows that the accessibility is significantly reduced in GAF_degradFP embryos at mitotically retained loci compared to interphase only or Zelda only (Figure 3c). Moreover, in the mitotically retained regions, TSS and enhancers but not TADs boundaries or other regions tend to have significantly reduced accessibility in GAF_degradFP embryos.

Reviewer #3

The authors address the role of GAF in mitotic bookmarking in the *Drosophila* embryo. They provide evidence for stable GAF binding and use mitotic ChIP to identify targets that retain GAF binding during mitosis. Subsets of these GAF targets have different chromatin marks. The authors use MS2 live imaging of the GAF mitotically retained target gene *scylla* in single nuclei to provide evidence that memory is disrupted upon GAF depletion. Overall, the data are of high quality but **one issue is that population level data on binding and chromatin modifications are correlated with single cell transcriptional responses, which makes interpreting the data more difficult.**

We thank Reviewer 3 for his/her feedback on our work.

We apologize if our description of the data in the original manuscript was unclear. Our intention was not to correlate single cell transcriptional responses to GAF binding by ChIP. We employed ChIP in order to identify relevant GAF-bound targets. We then monitored the timing of transcriptional activation and a potential memory bias with single nuclei live imaging of a subset of these GAF-bound loci. We attempted to connect these two types of data in a phenomenological descriptive model in the discussion, lines 365-376.

Major comments

3.1 *It is not clear to me exactly what the authors model is. I think it is that GAF can be bound pre/during mitosis and facilitate chromatin changes allowing rapid activation that is attributed to transcriptional memory. But what do the authors think is happening in the nuclei that become active from inactive mothers? Is it that they assume GAF is bound in the interphase to allow activation (with a lag) or is it that the gene of interest (eg *scylla*) is activated in the absence of GAF binding? I think the data presented are more supportive of the former. The results section ends with a line about stochastic GAF binding but this is never elaborated on. It would help the reader to discuss more fully what the data show and what the shortcomings are in the discussion.*

We agree that the original version of the discussion did not elaborate sufficiently on what our working model could be. We have now amended the text accordingly line 316. However we would like to clearly state that this model is a hypothetical one, as we can't directly quantify GAF binding in 'active mothers' versus 'inactive mothers'. We favor indeed the reviewers's first interpretation, that GAF binding may be the basis of a differential activation in nc13 (the mothers).

We amended our text line 316 as such: " We speculate that, during interphase of nc13, there would be a differential probability of GAF binding between active and inactive mother nuclei. This differential in GAF binding in interphase of nc13 would persist during mitosis (our data suggest that GAF residence time is long) and would explain why descendants of active nuclei, can activate transcription faster than those coming from inactive (GAF-unbound during mitosis) nuclei."

In the following comments 3.2 to 3.6, Reviewer 3 addresses concerns/questions regarding GAF nuclear puncta. As noted by the referee himself (and we totally agree), ' *Gaf puncta seem interesting but are barely discussed* '.

Indeed to our mind, what the hub/foci are doing is a paper in itself and is out of the scope of our manuscript centered around mitotic memory. Moreover, a rigorous analysis of GAF hubs would require many more GAF alleles. Therefore, while we attempted to answer below to some of the reviewer's questions, we cannot provide a full description of GAF hub dynamics.

3.2 *From the text description, it seems that the conclusion from the data shown in Sup Fig 1b*

is that the large GAF puncta are due to heterochromatin binding apically and the more basal uniform GAF signal represents binding to euchromatin. Is this right?

We believe that the large GAF puncta (Figure Sup 1b) located apically correspond to GAF protein within heterochromatin. Indeed, at this stage of embryonic development, nuclei are organized with centromeres located apically⁶. To reinforce this conclusion, we provide an image of n.c. 14 nuclei stained with GAF antibody and an antibody against the heterochromatin histone mark H3K9me2-3. This image shows co-localization between big GAF foci and H3K9me2-3 staining (Supplementary Figure 1c).

3.3 *In Fig S1 at nc14 there appears to be poor colocalization between the peri-centromeric DNA regions tested and GAF. Could it not be that the GAF puncta are associated with transcriptional activation as in a hub model?*

We agree with the reviewer that the peri-centromeric DNA regions tested do not obviously co-localize with GAF puncta. Given that the regions labeled are not (GAGA)_n but (AATAAACATAG)_n repeats (10bp satellite⁷), this result is quite expected. We employed this labeling to show that peri-centromere were located in the vicinity of large GAF puncta^{8,9}.

As suggested by the referee, we explored whether GAF nuclear hubs would be associated with transcriptional activation. In apical regions of wild-type nuclei, we did not detect an overlap between Pol II-Ser5P signal and GAF hubs (Figure for Referee 4). The data rather suggest an exclusion between active sites of transcription (Ser5P) and GAF apical puncta. Together with results of point 3.2 (Supplementary Figure 1c), these data confirm that GAF large nuclear puncta located apically primarily correspond to heterochromatin.

FIGURE 4: *Maximum intensity projected Z-stack images from immunofluorescence of PolII-Ser5 (pink) and GAF (green) on late n.c.14 wild type embryos. Nuclei are imaged on sagittal view. Scale bars are 5µm.*

3.4 *The GAF puncta seem interesting but are barely discussed.*

We agree, that this is a fascinating line of inquiry, but as previously noted above, we feel it is beyond the scope of this particular manuscript.

3.5 *The conclusion that there is stable binding of GAF to chromatin, including during mitosis,*

is central to the paper, yet the supporting evidence is weak. What is the evidence that the short and long recovery times from the FRAP experiment (Fig 1d) correspond to diffusion and stable binding, respectively?

Studying a GAF DNA binding mutant would strengthen this conclusion.

We totally agree that the interpretation of FRAP recovery curves is generally strengthened by FRAP experiments on DNA binding mutants. Unfortunately, our attempts to obtain GAF DNA binding mutant CRISPR lines have failed so far.

Having said that, we think that our evidence supporting GAF stable binding is not weak.

Indeed, for a DNA binding protein within the nucleus, kinetics in the range of minutes are unlikely to be reflecting a free diffusion. To verify this, we estimated GAF diffusion by performing FCS in the cytoplasm where no DNA should be bound by GAF (new Supplementary Figure 1d, g-h) and estimated two diffusion characteristic times, in the range of 20 $\mu\text{m}^2/\text{s}$ and 1 $\mu\text{m}^2/\text{s}$. These two diffusion times correspond to the two measured in the nucleoplasm (Supplementary Figure 1g and h), with the slow diffusion time being slightly slower in the nucleoplasm. We suspect that the fast diffusion time corresponds to free diffusing GAF-GFP molecules and the slow diffusion time to a GAF-specific effective diffusion coefficient caused by very transient unspecific binding as it has been shown to be the case for other TF¹⁰.

Importantly, this slow effective diffusion coefficient observed in FCS is in the same range as the one measured using FRAP, when analyzed with a diffusion-reaction model. We would note here that fast value of the diffusion coefficient is not experimentally accessible with FRAP). This strengthened our conclusion that the second time observed in the recovery curve correspond to specific binding.

We apologize if our description of these kinetics was unclear, and have now amended our text to describe more precisely our interpretation of FRAP and FCS data in the main text (lines 93-116).

In addition, during the course of this revision, work from Carl Wu's lab describes GAF binding kinetics with various mutants in *Drosophila* hemocyte cells¹¹. In this cellular context using single particle tracking, the authors retrieved a surprisingly long GAF residence time, on the order of minutes (85 sec). Therefore, their independent results are in total agreement with GAF kinetics that we retrieved in the blastoderm embryo. Of note, in both cases (our and their study), the analysis was based on the GAF short isoform which represents the predominant form in the early embryo. In hemocytes, using a DNA binding mutant allele, Tang et al., validated that GAF long residence time corresponded to site-specific DNA binding.

Collectively, these data, as well as our FRAP and FCS results in the embryo strongly support stable GAF binding to chromatin *in vivo*.

3.6 Were the FRAP and FCS measurements taken apically or basally?

FRAP and FCS measurements were taken in the middle of the nuclei as mentioned line 104, Figure 1c and new Supplementary Figure 1d.

It would be interesting to measure the stability of GAF DNA interaction in heterochromatin vs euchromatin, if that really is what is represented apically and basally, as there may be differential stability. I am also confused as to how the data in Fig 1e relates to Fig S1e, f.

Please see common response to comments 3.2 to 3.6, above. We apologize for the link between figures being misleading, we now provided a more in-depth description of the FRAP and FCS data in the main text and added supplementary figures.

While we acknowledge that decoding GAF binding kinetics in heterochromatin versus euchromatin would be of great interest, we believe that it is out of the scope of this manuscript.

3.7 In Fig 3, clusters 2 and 3 seem to be transcriptionally inactive – can the authors speculate what these subpopulations of repressed genes might represent?

Using published RNA-seq ¹², we examined transcriptional levels in these three clusters. In contrast to GAF targets of cluster 1, genes of cluster 2 and 3 mostly correspond to inactive genes (Figure for Referee 5).

FIGURE 5: Histograms of mean RNAseq signal in the 3 different clusters of GAF mitotically retained loci. Right panel represent the percentage of the ON and OFF genes in the three categories. A gene was considered ON above 5 RPKM detected.

As GAF is known to be recruited at Polycomb response elements, and as cluster 2 is characterized by H3K27me3 mark, we suspected it could be enriched in Pc. Indeed, this is the case as shown in Figure 3d and now commented lines 207-211.

We then searched for other known TF present at these early stages: Zelda ¹³, CBP ¹⁴, Dorsal ¹⁵, Opa ¹⁶ and Clamp ¹⁷ and examined their enrichment in the different clusters. However, no specificity for each of the cluster was found (Figure for referee 6). Interestingly, cluster 3 seems to be devoid of any TFs studied here. We can speculate that this cluster would represent non-transcribed regions, maybe structural regions such as insulators, TAD boundaries or anchored regions as GAF has been shown to be able to mediate DNA contact in *trans* ¹⁸.

FIGURE 6: Transcription factor enrichment in clusters of mitotically retained GAF peaks. Heatmaps of k-means clustered mitotically retained GAF peaks, based on Zelda, CBP, Dorsal, Opa and CLAMP ChIP-seq. n: number of identified GAF peaks.

CBP ChIP-seq	2-4h AEL	Koenecke et al., 2016	GSE68983
Zld ChIP-seq	n.c.8 , n.c.13, n.c.14	Harrison et al., 2011	GSE30757
DI ChIP-seq	2-3h AEL	Sun et al., 2015	GSE65441
Opa ChIP-seq	nc13-nc14	Koromila et al., 2020	GSE153329
CLAMP ChIP-seq	nc14	Rieder et al., 2019	GSE133637

Also, the embryo timings for the data shown in Fig 3/Fig S3 are not clear.

Embryo timings are now specified in the legends, for each panel of Figure 3 and Supplementary Figure 3.

3.8 *The MS2 data (Fig 4/Fig S6) show that only a proportion of nuclei are active in nc13 and nc14. Is it the case that typically only one daughter is active following division? I think the data in Fig S6e shows that at most 50% of the daughters from nc 13 active nuclei are also active in nc14 (but I may have misunderstood it, I found the legend confusing and it is not clear if the data relate to dorsal or ventral nuclei). Is there any difference in the ability of both daughters to activate transcription depending on whether the mother was active or inactive? Linked to this, it would be interesting if the authors could explain how they think bookmarking works molecularly – is it that both replicated chromosomes have GAF bound and modified chromatin or only one?*

As we are interested in the first activation after mitosis (in order to extract the waiting time used for mathematical modeling), we only quantify the timing of activation of the first activated daughter after mitosis. This is now specified in the figure legend, and we modified the graph of the instantaneous percentage of activation to only show the kinetics of ‘first daughters’ (Supplementary Figure 6e). Following the referee’s suggestion, we examined if the second daughters also show a transcriptional memory bias. The results are shown in Figure for Referee 7. When considering the population of second daughters, the transcriptional timing bias is still present. We therefore conclude that there is also a transcriptional memory bias in the second daughters.

FIGURE 7: Box plot representing the mean time of the activation after mitosis of nuclei derived from active (green) and inactive (purple) nuclei in scylla 24X-MS2 CRISPR/+ embryos of the first activated daughter and the second activated daughter. Centered horizontal line represents the median. Two-tailed Welch's t-test **** $p < 0.0001$.

3.9 GAF depletion slows the activation in daughter nuclei from both active and inactive nuclei. Does the statistically significant delay in activation observed in daughters from inactive mother nuclei in the presence of GAF RNAi (relative to inactive mothers with white RNAi) not mean that GAF has a memory independent role in activation? The key question is whether the similar activation time observed in daughters from active GAF RNAi and inactive white RNAi mothers is due to a loss of memory in the active mothers or simply a coincidence based on overall slower activation profiles. I did not find the argument that the effect of GAF RNAi on daughters from active mothers is due to a loss of memory completely convincing.

Here the referee questions the significance of the memory loss in GAF_RNAi and asks whether the memory loss is a coincidence.

In this manuscript and previous articles ^{1,19,20} our operational definition of memory is based on the difference of activation times in nuclei whose history is different (originating from active and inactive mothers). Thus, equality of activation times irrespective of history indicates a loss of memory. A coincidence is of course possible but highly unlikely, because it would involve several independent but exactly cancelling effects. In Ferraro et al., we tested whether the memory bias observed was different from a random, memoryless scenario and we found that it was indeed the case. Additionally, in our previous study of the pioneer factor Zelda, we found that slower activation profiles have the opposite effect, which is to reinforce memory. In general, a memory device (biological or electronic) needs a slow process that maintains the same state for a long time, such that the memory length is the timescale of the involved process.

Moreover, Dufourt et al.¹ demonstrated that Zelda depletion leads to a decrease in the tempo of transcription activation of the snail transgene, but a memory bias was still present between nuclei coming from active mothers and those coming from inactive mothers. Therefore transcriptional activation can be slowed-down without affecting the memory bias.

For these reasons, GAF-associated memory should not be influenced by the slowness of the transitions.

While we strongly believe that the reduction of memory observed in *GAF RNAi* embryos is not a coincidence, we would like to nuance our conclusion. We cannot prove that GAF is directly responsible for the memory reduction. Moreover, GAF depletion is not complete (as also quantified from WB experiments in new panel Sup 7b), so consequences on memory are not a complete memory erasure. We amended the text in this direction line 289.

3.10 *What do the authors think is the basis for the stochastic activation from inactive mothers? Could it be that GAF is only bound in a subset of the cells post-mitosis and for those without bound GAF the delay reflects a requirement for GAF recruitment? If this is correct, activation in daughters from inactive (but not active) mothers might depend on a high local GAF concentration, for example proximity to the GAF puncta. This should be testable by imaging GAF-GFP and MS2 (with MCP-RFP) simultaneously in nuclei.*

See response to point 3.1. We believe (but have yet to be definitively proved) that the stochastic activation in nc13 is due to a stochastic binding of GAF in nc13 nuclei. This GAF binding in nc13 (and in mitosis) would accelerate the timing required to activate transcription post-mitosis. In comparison, descendants of inactive nuclei will require more time for activation, as they would need to indeed recruit GAF in addition to the other steps prior to transcription activation.

In an attempt to describe the location of transcription activation of *scylla* (with respect to GAF concentration) we performed immuno-FISH with GAF antibody and MS2 probes on *scylla_24XMS2* embryos. These data are shown in a new additional panel in Supplementary Figure 6g. While GAF large puncta are located apically, MS2 transcription foci are not overlapping and are located in the middle of nuclear space.

However, we can't exclude that a subset of MS2 foci might colocalize with smaller GAF foci. These smaller size GAF foci are difficult to image in living embryos as they are highly mobile. As GAF large hubs don't co-localize with sites of transcription of *scylla*, we did not perform the time-consuming dual-color GAF and MS2/MCP-RFP τ imaging; it would also have required generating a strain with a histone fused to other fluorescent tag than GFP or RFP with enough signal to be detected for segmentation by our software.

3.11 *Can the authors speculate as to what they think the different states represent in their model?*

In the main manuscript, we refrained from interpreting the nature of the states of the modeling as this model is purely phenomenological. An important aspect to consider is that this modeling framework is solely based on timing to first activation and does not consider the second-scale fluctuations in signal intensities. Thus, we are not revealing the rate limiting steps occurring at steady state responsible for promoter bursting. In a recent study, we have shown that the distribution of post-mitotic waiting time operates in a very distinct temporal regime than those implicated in promoter bursting²¹.

The competent ON state could correspond to a promoter for which the PIC is assembled. The nature of the multiple OFF states is more difficult to interpret. But one such state could correspond to nucleosome eviction, which duration would vary in the presence or absence of GAF. Another OFF state could potentially correspond to the recruitment of chromatin remodelers such as PBAP, Nurf or FACT. As this is highly speculative, we prefer not to discuss these plausible interpretations in the main text.

Minor comments

How do the authors know that the GAF-GFP being measured in the nucleus is on DNA in the interphases, as indicated in Fig 1b?

We agree with this comment. We have now replaced ‘on DNA’ by ‘in nucleus’ in the panel Figure 1b.

In Fig 2f – what does the ‘other’ category for the mitosis only class reflect? Is it mostly gene bodies or intergenic regions?

This category indeed primarily includes intergenic regions. This is now specified in the figure legend of Fig2f. This is confirmed with the new analysis with HOMER tool in Supplementary Figure 2e.

Fig S3a shows that a higher number of GAGAG motifs is associated with a wider region of more weakly accessible chromatin. What might the reason for this be?

As a metaprofile visualization is hard to interpret with intensities, we provide a Figure of accessibility profile depending on the number of GAGAG repeats (Figure for referee 8).

FIGURE 8: Chromatin accessibility at regions with different number of GAGAG sites. Metagene profiles of the ATAC-seq signal in interphase (dark blue) and mitosis (light blue) at GAF mitotically retained regions, partitioned by the number of GAGAG motifs they contain.

We do see a tendency of ‘widening’ of accessibility regarding the number of GAGAG motifs, but this is likely due to the property of GAF to open chromatin; by extension, the more GAF protein is able to bind, the more the chromatin will be accessible.

For the cartoon in Fig 4d – should the green nucleus at nc13 not be depicted as transcriptionally active?

Thanks for pointing out this mistake, this is now corrected.

References

1. Dufourt, J. *et al.* Temporal control of gene expression by the pioneer factor Zelda through transient interactions in hubs. *Nat. Commun.* **9**, 1–13 (2018).
2. Blythe, S. A. & Wieschaus, E. F. Establishment and maintenance of heritable chromatin structure during early drosophila embryogenesis. *Elife* **5**, 1–21 (2016).
3. Gaskill, M. M., Gibson, T. J. ., Larson, E. D. . & Harrison, M. M. . GAF is essential for zygotic genome activation and chromatin accessibility in the early Drosophila embryo. *Elife* 0–43 (2021).
4. Cardozo Gizzi, A. M. *et al.* Microscopy-Based Chromosome Conformation Capture Enables Simultaneous Visualization of Genome Organization and Transcription in Intact Organisms. *Mol. Cell* **74**, 212-222.e5 (2019).
5. Ghavi-Helm, Y. *et al.* Enhancer loops appear stable during development and are associated with paused polymerase. *Nature* **512**, 96–100 (2014).
6. Wilkie, G. S., Shermoen, A. W., O’Farrell, P. H. & Davis, I. Transcribed genes are localized according to chromosomal position within polarized Drosophila embryonic nuclei. *Curr. Biol.* **9**, 1263–1266 (1999).
7. Garavís, M. *et al.* The structure of an endogenous Drosophila centromere reveals the prevalence of tandemly repeated sequences able to form i-motifs. *Sci. Rep.* **5**, 1–10

- (2015).
8. Lohe, A. R., Hilliker, A. J. & Roberts, P. A. Mapping simple repeated DNA sequences in heterochromatin of *Drosophila melanogaster*. *Genetics* **134**, 1149–1174 (1993).
 9. Raff, J. W., Kellum, R. & Alberts, B. The *Drosophila* GAGA transcription factor is associated with specific regions of heterochromatin throughout the cell cycle. *EMBO J.* **13**, 5977–5983 (1994).
 10. Elf, J., Li, G.-W. & Xie, X. S. Probing Transcription Factor Dynamics at the Single-Molecule Level in a Living Cell. *Science (80-.)*. **316**, 1191 LP – 1194 (2007).
 11. Tang, X. *et al.* Kinetic principles underlying pioneer function of GAGA transcription factor in live cells. *bioRxiv* (2021) doi:10.1101/2021.10.21.465351.
 12. Lott, S. E. *et al.* Noncanonical compensation of zygotic X transcription in early *Drosophila melanogaster* development revealed through single-embryo RNA-Seq. *PLoS Biol.* **9**, (2011).
 13. Harrison, M. M., Li, X. Y., Kaplan, T., Botchan, M. R. & Eisen, M. B. Zelda binding in the early *Drosophila melanogaster* embryo marks regions subsequently activated at the maternal-to-zygotic transition. *PLoS Genet.* **7**, (2011).
 14. Koenecke, N., Johnston, J., Gaertner, B., Natarajan, M. & Zeitlinger, J. Genome-wide identification of *Drosophila* dorso-ventral enhancers by differential histone acetylation analysis. *Genome Biol.* **17**, 1–19 (2016).
 15. Sun, Y. *et al.* Zelda overcomes the high intrinsic nucleosome barrier at enhancers during *Drosophila* zygotic genome activation. *Genome Res.* **25**, 1703–1714 (2015).
 16. Koromila, T. *et al.* Odd-paired is a pioneer-like factor that coordinates with zelda to control gene expression in embryos. *Elife* **9**, 1–71 (2020).
 17. Rieder, L. E., Jordan, W. T. & Larschan, E. N. Targeting of the Dosage-Compensated Male X-Chromosome during Early *Drosophila* Development. *Cell Rep.* **29**, 4268-4275.e2 (2019).
 18. Mahmoudi, T., Katsani, K. R. & Verrijzer, C. P. GAGA can mediate enhancer function in trans by linking two separate DNA molecules. *EMBO J.* **21**, 1775–1781 (2002).
 19. Ferraro, T. *et al.* Transcriptional Memory in the *Drosophila* Embryo. *Curr. Biol.* **26**, 212–218 (2016).
 20. Bellec, M., Radulescu, O. & Lagha, M. Remembering the past: Mitotic bookmarking in a developing embryo. *Curr. Opin. Syst. Biol.* **11**, 41–49 (2018).
 21. Pimmett, V. L. *et al.* Quantitative imaging of transcription in living *Drosophila* embryos reveals the impact of core promoter motifs on promoter state dynamics. *Nat. Commun.* **12**, 4504 (2021).

Reviewers' Comments:

Reviewer #1:

Remarks to the Author:

The authors have addressed all of my concerns.

Reviewer #2:

Remarks to the Author:

Thanks to the authors for the revision. All my concerns related to data analysis have been addressed.

Reviewer #3:

Remarks to the Author:

The manuscript is significantly improved and overall I am happy that the authors have addressed my concerns. However, I feel that the description of the FCS and FRAP in the main text is very confusing. I had to read it a number of times to understand what I think the authors are meaning. I found this line from the rebuttal helpful – “We would note here that fast value of the diffusion coefficient is not experimentally accessible with FRAP” – but it is absent from the manuscript text. I suggest that the authors might like to check whether the paragraphs describing the FCS and FRAP (in particular the logic behind the conclusions) can be modified to be more reader friendly.

Also, as a very minor point, I do not think that the cartoon in Fig 4d has been fixed as indicated in the rebuttal (see my last original point).

Point by point answer to reviewer's comments

The control of transcriptional memory by stable mitotic bookmarking

Maëlle Bellec¹, Jérémy Dufourt¹, George Hunt², Hélène Lenden-Hasse¹, Antonio Trullo¹, Amal Zine El Aabidine¹, Marie Lamarque¹, Marissa M Gaskill³, Heloïse Faure-Gautron¹, Mattias Mannervik², Melissa M Harrison³, Jean-Christophe Andrau¹, Cyril Favard⁴, Ovidiu Radulescu⁵, and Mounia Lagha¹.

Our answers are in blue.

REVIEWERS' COMMENTS

Reviewer #1 (Remarks to the Author):

The authors have addressed all of my concerns.

We thank the reviewer #1 for all the comments helping to improve the manuscript.

Reviewer #2 (Remarks to the Author):

Thanks to the authors for the revision. All my concerns related to data analysis have been addressed.

We thank the reviewer #2 for the remarks improving the manuscript.

Reviewer #3 (Remarks to the Author):

The manuscript is significantly improved and overall I am happy that the authors have addressed my concerns. However, I feel that the description of the FCS and FRAP in the main text is very confusing. I had to read it a number of times to understand what I think the authors are meaning. I found this line from the rebuttal helpful – “We would note here that fast value of the diffusion coefficient is not experimentally accessible with FRAP” – but it is absent from the manuscript text. I suggest that the authors might like to check whether the paragraphs describing the FCS and FRAP (in particular the logic behind the conclusions) can be modified to be more reader friendly.

We thank the reviewer for this positive answer. We modified the main text regarding the comment :

« We would note here that fast value of the diffusion coefficient observed with FCS is not experimentally accessible with our FRAP device. A possible interpretation of these two kinetic timescales observed with FRAP experiments, would be that the fast residence time corresponds to GAF non-specific binding as observed previously 7, while the long-lived residence time would correspond to sequence-specific binding to its consensus binding sites. »

Also, as a very minor point, I do not think that the cartoon in Fig 4d has been fixed as indicated in the rebuttal (see my last original point).

Indeed, this has now been corrected, thanks.